# Multilayered regulations of alternative splicing, NMD, and protein stability control temporal induction and tissue-specific expression of TRIM46 during axon formation

John K. Vuong[1], Volkan Ergin[1], Liang Chen[2] & Sika Zheng [1,3✉]

The gene regulation underlying axon formation and its exclusiveness to neurons remains elusive. TRIM46 is postulated to determine axonal fate. We show *Trim46* mRNA is expressed before axonogenesis, but TRIM46 protein level is inhibited by alternative splicing of two cassette exons coupled separately to stability controls of *Trim46* mRNA and proteins, effectively inducing functional knockout of TRIM46 proteins. Exon 8 inclusion causes nonsense-mediated mRNA decay of *Trim46* transcripts. PTBP2-mediated exon 10 skipping produces transcripts encoding unstable TRIM46 proteins. During axonogenesis, transcriptional activation, decreased exon 8 inclusion, and enhanced exon 10 inclusion converge to increase TRIM46 proteins, leading to its neural-specific expression. Genetic deletion of these exons alters TRIM46 protein levels and shows TRIM46 is instructive though not always required for AnkG localization nor a determinant of AnkG density. Therefore, two concurrently but independently regulated alternative exons orchestrate the temporal induction and tissue-specific expression of TRIM46 proteins to mediate axon formation.

[1] Division of Biomedical Sciences, School of Medicine, University of California, Riverside, Riverside, CA 92521, USA. [2] Department of Quantitative and Computational Biology, University of Southern California, Los Angeles, CA 90089, USA. [3] Center for RNA Biology and Medicine, University of California Riverside, Riverside, CA 91521, USA. ✉email: sika.zheng@ucr.edu

Neurons are highly polarized, containing an axon that conducts action potentials to directionally transmit information between neurons. An axon is specified during neuronal differentiation and is a prerequisite to synapse and neural circuit formation. Newly generated immature neurons typically contain multiple seemingly indistinguishable neurites. One neurite extends its growth and is specified for an axonal fate, leading to the establishment of axonal and somatodendritic compartments. Axon formation appears to involve a diverse set of molecules, including growth factors, signal transducers, kinases, phosphatases, cytoskeletal proteins, and molecular motors[1–12]. Substantial progress has been made toward understanding the cellular processes mediated by these molecules, although the gene expression regulation underlying these processes remains elusive.

The search for axon determinants has focused on symmetry-breaking molecules exhibiting differential localization at the future axon vs other neurites. Localization of kinesin motor proteins (e.g., KIF5C, KIF1A, and KIF3A) in the nascent axon plays an essential role in establishing the axonal domain by promoting polarized cargo trafficking[6,13,14]. Other proteins separating neuronal compartments include widely used axonal marker Tau1 and dendritic marker microtubule-associated protein 2 (MAP2)[15]. However, knockout (KO) of *Tau1* or *Map2* in mice show no obvious effects on brain development or neuronal polarity, probably due to redundant functions of other MAPs such as MAP1B[16–18]. Many kinesin knockouts show diverse phenotypes in mice, commonly disrupting cargo delivery and reducing axonal transport rates. However, such mice do not exhibit clear developmental defects of axon formation either[19–21].

Amidst the multi-decade chase for the earliest molecular determinants of axon specification, tripartite-motif containing 46 (TRIM46) has emerged as another promising candidate. TRIM46 was recently shown to localize to future axons earlier than Tau1, suggesting TRIM46 is one of the earliest, if not the earliest, markers for axon specification[4]. TRIM46 localization at the proximal axon partly overlaps with the future axon initial segment (AIS). The AIS is a distinct neuronal compartment at the base of the axon best known for the generation of the action potential and regulation of cargo trafficking[9,18,19]. AIS is marked by the accumulation of AnkG, which is essential to the maintenance of axo-dendritic polarity. It has been proposed that TRIM46 is transported under low MARK2 activity to the future AIS where TRIM46 organizes microtubule fascicles leading to enrichment of AnkG at the AIS site[4,22–24].

While regulation of TRIM46 subcellular localization is a key component influencing axonal fate, temporal control of TRIM46 expression likely serves another important checkpoint but this remains to be determined. Axon consistently emerges during a specific developmental time window, arguing for the importance of temporal regulation. Furthermore, neural-specific expression of axonal determinants could be a parsimonious explanation for neurons' exclusive ability to generate an axon. However, gene expression controls of molecular candidates, including *Trim46*, have rarely been examined. Finally, despite accumulating evidence for TRIM46's important role, the effect of a complete depletion of TRIM46 is unknown (e.g., whether TRIM46 is always required for AnkG localization). This knowledge is critical to guide the future direction of research.

Alternative splicing of precursor mRNA produces mRNA isoforms from a single gene and is an essential regulatory mechanism of gene expression. Mammalian brains exhibit prevalent splicing control governed by RNA binding proteins (RBPs)[25–31]. Early cortical axonogenesis is controlled by a neural-specific alternative splicing program coordinated by polypyrimidine tract binding protein 2 (PTBP2)[32]. The RBFOX family proteins control alternative splicing of many genes (but

not *Trim46*) encoding the AIS components to regulate AIS formation[33]. Axon guidance of dorsal interneurons in the spinal cord is controlled by NOVA proteins, which regulate isoform expression of attractive DCC and repulsive ROBO receptors of growing commissural axons[34–36]. Mutation of these RBPs show detrimental defects in brain development, likely resulting from mis-regulation of a large set of alternative exons. However, the biological significance of most individual alternative splicing events and their possible contribution to axon formation are largely unknown[28].

The nonsense-mediated mRNA decay pathway (NMD) functions as a mRNA quality control mechanism that selectively degrades transcripts containing a premature termination codon (PTC), preventing translation of abnormal C-terminal truncated proteins[37–39]. While NMD is classically recognized as a surveillance pathway, it is increasingly viewed as a regulatory process with an active role in fine-tuning the abundance of normal transcripts[40–44]. In mammalian cells, alternative splicing coupled with nonsense-mediated decay (AS-NMD) is a conserved mechanism of post-transcriptional gene regulation. A common NMD-inducing feature is a PTC at least 50 nt upstream of an exon-exon junction[37–39]. Normal stop codons are typically found in the last exon downstream of any exon-exon junction. Alternative splicing can alter the reading frame to produce a PTC-containing spliced transcript subject to NMD[31,45–48]. By shifting the splicing of NMD-associated alternative exons, AS-NMD effectively controls homeostatic expression of multiple splicing factors[49–54]. AS-NMD also regulates the expression of developmental genes in the brain (e.g., *Psd-95*, *Bak1*, and histone modifiers)[40,46,49,55]. For *Psd-95* and *Bak1*, the NMD-associated exons exhibit tissue-specific splicing, resulting in neural-specific expression of the PSD-95 protein and neural-specific loss of the BAK1 protein. Therefore, AS-NMD can play an important role in regulating the overall abundance of a functional protein during neural development.

In this study, we investigated whether and how *Trim46* expression is developmentally regulated during neuronal differentiation to control axon formation. We found *Trim46* expression is increased in concordance with axonogenesis. Surprisingly, *Trim46* upregulation is orchestrated by alternative splicing of its exon 8 and exon 10 in addition to transcriptional upregulation. Exons 8 and 10 are independently regulated and lead to different downstream control of *Trim46* expression, but resulting in convergent upregulation of the functional TRIM46 protein to enable axon formation. Using CRISPR/Cas9-mediated exon knockout, we show alternative splicing controls of exons 8 and 10 are functional for TRIM46 induction and AnkG recruitment to the proximal axon. Therefore, multiple control mechanisms act synergistically to determine the temporal induction of TRIM46 during early axon formation.

## Results

**Trim46 exon 10 is developmentally regulated**. The success in differentiating embryonic and induced pluripotent stem cells into functional neurons in the absence of an in vivo microenvironment presents a unique system for study of the temporal control of axon formation. We differentiated the well-established 46C Sox1-GFP knock-in mouse embryonic stem cells (ESC) into neurons (Figs. 1a and 1b)[56–59]. The GFP expression under the control of the Sox1 promoter provides a convenient readout of successful differentiation toward the neural lineage and of obtaining neural progenitors. Briefly, ESCs were dissociated in single cells and replated in 96-well u-bottom and agarose-coated low cell adhesion plates. With differentiation medium, these cells reaggregate over time and form embryoid bodies (EB) resembling

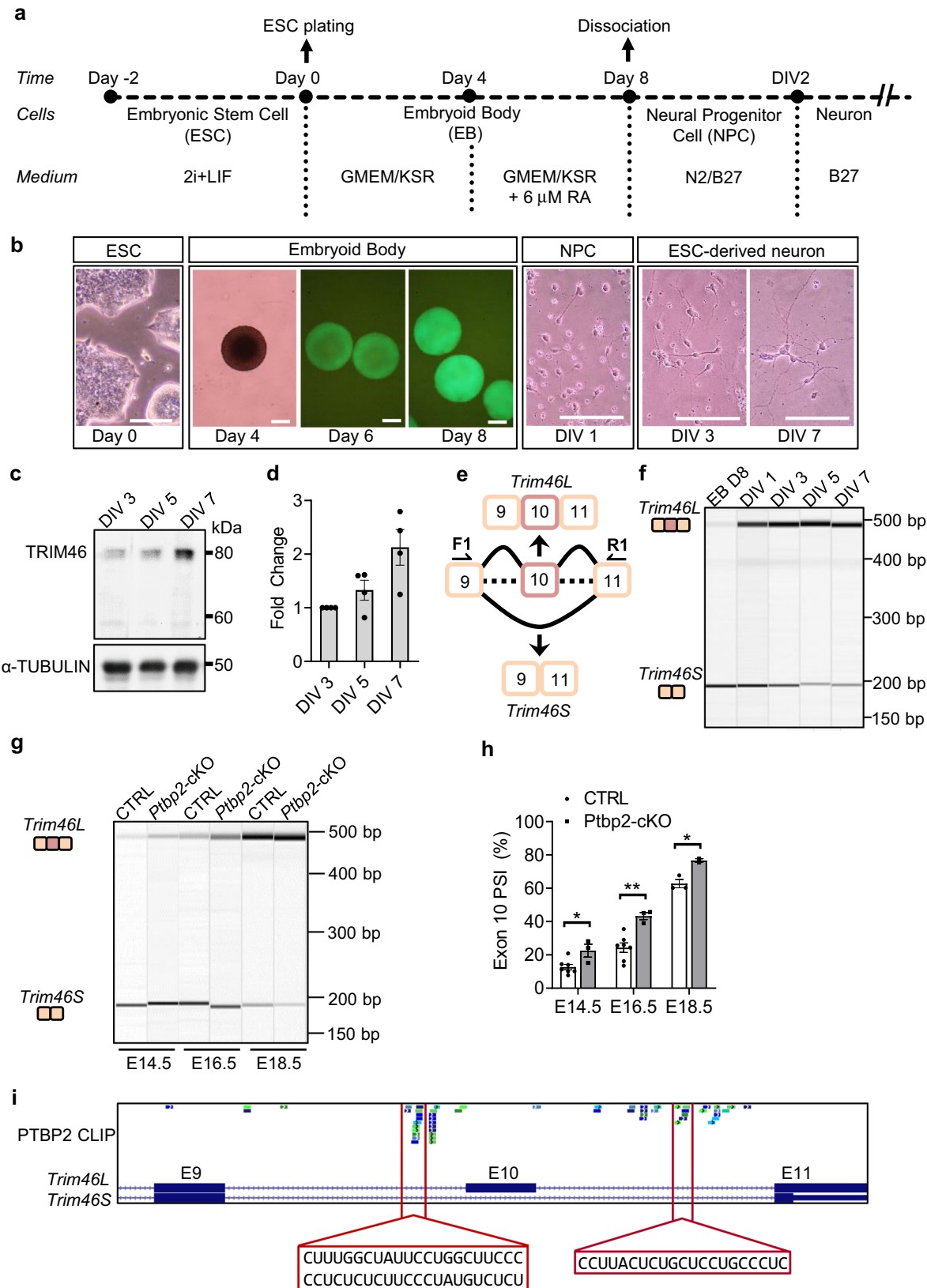

ectoderm cells[57,60]. On day 4, retinoic acid (RA) is added to activate neurogenic pathways, and the induction of neural lineages is highlighted by increasing GFP expression from day 6 to day 8 (Fig. 1b). On day 8, Sox1-GFP expressing EBs were dissociated and plated as a monolayer (Figs. 1a and 1b). EB-derived cells became neural progenitor cells (NPC) on day

in vitro 1 (DIV 1)[56–58,61]. Further differentiation led to the polarization of post-mitotic neurons around DIV 2/3. Many neurons display a singular long cellular process (the future axon) and other short processes (the future dendrites) on DIV 3. By DIV 7, differentiated cells exhibited a typical neuronal morphology with elaborated dendrites and a very long axon

**Fig. 1 *Trim46* exon 10 is developmentally regulated. a** Schematic protocol for embryoid body (EB) based neuronal differentiation using 46C embryonic stem cells (ESC). Differentiation medium included Glasgow's modification of Eagle's Minimal Essential Medium (GMEM) and Knockout Serum Replacement (KSR) for EB day 0 to day 8 while retinoic acid (RA) was used from EB day 4 to day 8. **b** Representative image of various differentiation stages from ESC to EB to neural progenitor cells (NPC), and into neurons from 45 independent experiments. Scale bar, 100 μm. **c** Western blot for TRIM46 protein expression from ESC-derived neurons on DIV 3, 5, and 7. **d** Quantification shows TRIM46 protein is upregulated during neuronal differentiation. Data represent mean ± SEM from 4 independent experiments. **e** Schematic showing mRNA isoforms produced from alternative splicing of *Trim46* exon 10. The inclusion of exon 10 represents the long isoform or (*Trim46L*) while the exclusion of exon 10 leads to shorter isoform (*Trim46S*). F1 and R1 show the location of primers for RT-PCR analysis of *Trim46* splicing. **f** Gel image shows isoform expression of *Trim46* mRNA in WT EBs on day 8 to ESC-derived neurons on DIV 7 during differentiation from 4 independent experiments. **g** Gel image highlighting alternative splicing of *Trim46* exon 10 between CTRL and *Ptbp2*-cKO mouse neocortices on E14.5, E16.5, and E18.5 from 3 independent experiments. **h** Splicing quantification shows upregulation of *Trim46* exon 10 PSI (percent spliced-in) in *Ptbp2*-cKO neocortices. Data is represented as mean ± SEM. Statistics: $t$-test, two-tailed, unpaired; (E14.5: CTRL, $n = 7$ and KO, $n = 3$; *$p = 0.02133$; E16.5: CTRL, $n = 7$ and KO, $n = 3$; **$p = 0.003421$; E18.5: CTRL, $n = 3$ and KO, $n = 2$; *$p = 0.02448$). **i** Genome browser track of E18.5 PTBP2 CLIP-Seq shows multiple distinct PTBP2 binding sites located in *Trim46* introns 9 and 10. Two prominent binding regions are highlighted.

demonstrated by axonal markers Tau1 and AnkG (Fig. 1b, Supplementary Fig. 1a).

Prior studies focused on localized TRIM46 activity at the prospective AIS site for axon specification[4]. To test whether the emergence of TRIM46 activity at the time of axon formation is a function of increased TRIM46 expression, we harvested protein lysates from multiple differentiation stages and performed Western blot analysis for TRIM46 protein. TRIM46 was not detected during the EB stage nor at DIV 1 (Supplementary Fig. 1b). On DIV 3, TRIM46 protein became detectable by Western Blot and was localized to the proximal axon (Fig. 1c, Supplementary Fig. 1b, 1c). Its expression further increased at DIV 5 and DIV 7 (Figs. 1c, 1d, and Supplementary Fig. 1b). Therefore, TRIM46 protein is not expressed in neural progenitors and is induced concurrent with the emergence of an axon.

We searched databases of NCBI RefSeq and Ensembl for annotated *Trim46* transcripts and found exon 8 and exon 10 are the two cassette exons consistently annotated in both databases. Importantly, skipping exon 10 causes a shift in the reading frame at the C-terminus, resulting in a much shorter isoform (TRIM46S) with an alternative stop codon in the last exon, exon 11 (Fig. 1e). We note that the TRIM46 protein reported in previous studies is the long isoform (TRIM46L) including exon 10 and TRIM46S is an uncharacterized isoform[4,22]. The large difference between TRIM46L (83 kDa) and TRIM46S (60 kDa) proteins suggests alternative splicing of exon 10 has functional implications.

To test whether exon 10 is alternatively spliced during neuronal differentiation we designed primers annealing to exon 9 and 11 to simultaneously detect the inclusion and skipping isoforms and performed quantitative alternative splicing analysis (Figs. 1e and 1f). We found EBs expressed *Trim46S* transcripts almost exclusively. During the EB-to-DIV 1 transition, the dominant *Trim46* mRNA isoform switched from *Trim46S* to *Trim46L*, and the trend continued until DIV 7 (Fig. 1f). Note that on Western blots, the upregulated TRIM46 protein is the long isoform mirroring the upregulation of the *Trim46L* mRNA. Therefore, the induction of the TRIM46 protein follows the mRNA isoform switch from *Trim46S* to *Trim46L*. Surprisingly, despite clear expression of *Trim46S* mRNA in EBs and on DIV 1, we could not detect the TRIM46S protein during these stages, indicating additional regulation to restrain TRIM46 protein expression.

The alternative splicing program associated with early axonogenesis is coordinated by PTBP2[32]. To test whether *Trim46* exon 10 is controlled by PTBP2, we depleted PTBP2 in embryonic forebrains using the Emx1-Cre line. We crossed *Ptbp2*[loxp/loxp] mice with *Ptbp2*[loxp/+] Emx1-cre mice to produce *Ptbp2*-conditional knockout mice (*Ptbp2*-cKO, or *Ptbp2*;[loxp/loxp] Emx1-cre) and control littermates (CTRL, *Ptbp2*[loxp/loxp])[32,62].

The major wave of neurogenesis in the mouse neocortex occurs during embryonic day 13 (E13) and E16. Axons emerge about 1–2 days after neurogenesis and continue to grow and mature during the embryonic period. We found exon 10 splicing increases from 12.7% in E14.5 to 62.7% in E18.5 WT cortices (Figs. 1g and 1h), consistent with the splicing switch during the ESC differentiation that generates the *Trim46L* isoform (Fig. 1f). This upward trend is also consistent with the reanalysis of cortex RNA-seq data from the ENCODE project (Supplementary Fig. 2a, 2b). In the *Ptbp2*-cKO cortices, exon 10 inclusion was consistently and significantly higher than in WT (22.5% vs 12.7% on E14.5, 43.3% vs 24.3% on E16.5, and 76.6% vs 62.7% on E18.5, Figs. 1g and 1h). These results demonstrate that loss of splicing regulator PTBP2 leads to an increase in the exon 10 + *Trim46L* variant during development.

In further support of PTBP2 regulating exon 10 splicing, we examined the PTBP2 CLIP-Seq dataset of E18.5 mouse brains[63]. PTBP2 typically binds flanking intronic sequences to regulate an alternative exon[62,63]. As shown in Fig. 1i, PTBP2 interacts with multiple polypyrimidine tracts within intron 9 and intron 10 of *Trim46*. These polypyrimidine tracts run from 11 to 24 nt and on average consist of 89% C and U nucleotides. There was also a strong PTBP2 CLIP-Seq signal in exon 11, represented by two notable CU-rich tracts from 9 to 24 nt (Supplementary Fig. 3a). Based on the PTBP2 CLIP-Seq data from embryonic mouse brains and analyses of embryonic PTBP2 conditional KO brains, we conclude PTBP2 represses the inclusion of *Trim46* exon 10.

**Presumptive endogenous TRIM46S protein isoform is undetectable.** Interestingly, both *Trim46L* and *Trim46S* mRNA transcripts were easily detected, but only the TRIM46L protein isoform was detected. We needed to rule out the absence of TRIM46S protein as an artifact of TRIM46 antibodies. We expressed N-terminal GFP-tagged TRIM46 protein isoforms (GFP-TRIM46L and GFP-TRIM46S) in N2a cells for immunoblot analysis. Two different TRIM46 antibodies (from the Hoogenraad lab and Proteintech Inc.) readily detected the ectopically expressed TRIM46S protein isoform (Supplementary Fig. 4a and 4b). By comparing the GFP signal to the TRIM46 signal, the Hoogenraad lab antibody was modestly less efficient in detecting TRIM46S, but the Proteintech antibody showed the same level of sensitivity to both isoforms. To further validate the antibody detecting TRIM46S, we ectopically expressed N-terminal FLAG-tagged TRIM46 proteins (FLAG-TRIM46L and FLAG-TRIM46S) and confirmed detection of both protein isoforms with the same sensitivity by the Proteintech TRIM46 antibody (Supplementary Fig. 4c). The Proteintech antibody was used in the remaining experiments. Our results demonstrated that the absence of endogenous TRIM46S protein was not due to detection sensitivity of TRIM46 antibodies.

We hypothesized that the lack of TRIM46S protein detection was due to the insolubility of TRIM46S in protein lysates. We overexpressed FLAG-TRIM46L and FLAG-TRIM46S proteins in N2a cells and compared TRIM46 expression in the supernatant versus the pellet fractions. We found most TRIM46L and TRIM46S proteins remained in the supernatant, as for either isoform only about 17% of TRIM46 proteins were in the pellet fraction (Supplementary Fig. 5a). Considering overexpression could enhance misfolding and insolubility of the ectopic proteins, endogenous TRIM46 proteins may have a lower insoluble proportion. Indeed, when we examined the pellet fractions from protein lysates of ESC-derived neurons, we observed no presumptive TRIM46S proteins by Western blot (Fig. 2a). These data suggest other causes for this absence of detection of the endogenous TRIM46S protein.

To enhance detection sensitivity, we attempted to enrich endogenous TRIM46S proteins by immunoprecipitation (IP). We first tested whether the TRIM46 antibody was suitable for IP using ectopically expressed TRIM46 proteins. We transfected N2a cells with plasmids expressing FLAG-Trim46L and FLAG-Trim46S cDNA and collected whole cell lysates 48 h after transfection. We used the TRIM46 antibody along with protein A magnetic beads in complex to pull down FLAG-TRIM46L or FLAG-TRIM46S proteins. FLAG-TRIM46L and FLAG-TRIM46S proteins were easily immunoprecipitated by the TRIM46 antibody but not the control IgG antibodies (Fig. 2b).

Next, we performed TRIM46-IP using P10 mouse cortices and E12.5 mouse brains. Endogenous TRIM46L protein was effectively immunoprecipitated and detected in P10 mouse cortices (Fig. 2c). TRIM46L proteins were also detected by TRIM46-IP in the E12.5 mouse brains but at a much lower level (Fig. 2d). In neither condition did we observe a protein band corresponding to the approximate 60 kDa size of TRIM46S protein. Importantly, E12.5 cortices expressed predominantly the Trim46S mRNA variant but only the TRIM46L protein isoform was detectable in this experiment.

Another possibility for the missing endogenous TRIM46S protein was that its epitope for the TRIM46 antibody was somehow masked (e.g., by post-translational modifications). This is unlikely: two different effective polyclonal TRIM46 antibodies failed to detect TRIM46S proteins under the denaturing conditions of SDS-PAGE. Nevertheless, epitope masking remained a possibility for the IP experiment. We decided to knock in a 3x-FLAG tag sequence to fuse in-frame at the N-terminal of the endogenous Trim46 locus and replace the start codon. This would create an allele endogenously expressing N-terminal FLAG-tagged TRIM46 protein and we would apply a commonly used FLAG antibody with high IP efficiency to detect and identify the uncharacterized short isoform.

We used a CRISPR/Cas9 strategy with a single-stranded oligo DNA (ssODN) donor template to take advantage of homology-dependent repair, and successfully generated the 3xFLAG-Trim46 knock-in allele (3xFKI:Trim46). Specifically, we designed a specific guide RNA targeting 5'UTR-exon 1 of Trim46 and a ssODN donor template with a 3xFLAG sequence. We transfected the gRNA, Cas9, and the ssODN donor into 46C ES cells (Figs. 2e and 2f)[64–67]. CRISPR/Cas9-edited stem cells were then selected and single cell-derived KI clones were isolated and screened for correct knock-in (Fig. 2f). Targeted insertion of 3xFLAG sequence to endogenous Trim46 locus was verified by PCR genotyping (Fig. 2g) and Sanger sequencing (Fig. 2h).

Western blot analysis using a FLAG antibody further confirmed the KI allele. However, using total protein lysates we only detected the 3xFLAG-TRIM46L protein from the lysate of neurons derived from 3xFKI:Trim46 ESC (Fig. 2i). We conducted FLAG-IP experiments using protein lysate from 3xFLAG-TRIM46-expressing neurons. Again, we were able to only identify

FLAG-TRIM46L proteins consistently. An approximate 60 kDa band that migrated slightly faster than the expected FLAG-TRIM46S proteins occasionally showed up but was inconsistent among biological replicates (Fig. 2j). Nevertheless, this band was dramatically weaker than FLAG-TRIM46L. In summary, we conclude that the endogenous TRIM46S protein is difficult to detect with existing methods.

**TRIM46S protein variant is less stable than TRIM46L protein.** We hypothesized protein stability might be different between the two protein isoforms. We tested this using ectopically expressed TRIM46 protein isoforms. The difference in protein stability can be reflected by steady-state abundance when protein translation is modest and at par with the protein decay level; overabundant translation can shadow protein degradation activity. We transfected increasing amounts of FLAG-Trim46L and FLAG-Trim46S cDNA plasmids (in the CAGIG backbone) in N2a cells and quantified protein isoform expression as a function of Trim46 plasmid amount. Backbone plasmid CAGIG was added to make the total transfection amount consistent and allowed EGFP as an internal expression control. At limiting dosages of 0.1, 0.2, and 0.4 μg of transfected plasmid, the FLAG-TRIM46S protein was expressed at a significantly lower level than the FLAG-TRIM46L isoform (Figs. 3a and 3b). There was no difference in expression levels with 1.0 μg DNA transfection, likely due to excess plasmid (Figs. 3a and 3b). Different protein outputs can be due to different mRNA levels of the two FLAG-tagged isoforms. To clarify this potential effect, we measured FLAG-Trim46L and FLAG-Trim46S transcripts at corresponding transfection dosages and found negligible difference of mRNA levels between isoforms (Supplementary Fig. 6a). These results indicate that the lack of TRIM46S protein detection in comparison to TRIM46L protein can be mimicked by expressing low amounts of exogenous TRIM46 proteins.

To further evaluate protein stability differences between the two isoforms, we performed a protein stability assay using translation inhibitor cycloheximide (CHX). N2a cells expressing FLAG-TRIM46L or FLAG-TRIM46S proteins were treated with CHX to inhibit protein synthesis and the remaining FLAG-TRIM46 protein isoforms were measured at subsequent time points. Western blot analysis showed FLAG-TRIM46 protein abundance decreased over a 24-h period after CHX treatment. FLAG-TRIM46 proteins were first normalized to GFP and then to starting levels of the first collection time point (0 h, i.e., right before CHX treatment). There were significant differences in the remaining protein proportions between FLAG-TRIM46L and FLAG-TRIM46S at 6, 12, and 24 h after CHX treatment (Figs. 3c and 3d). For example, at 12 h, 31% FLAG-TRIM46L protein and 6% FLAG-TRIM46S protein remained (Figs. 3c and 3d).

The difference was also obvious when we co-transfected FLAG-Trim46L and FLAG-Trim46S cDNA plasmids together and detected the remaining protein isoforms after CHX treatment (Supplementary Fig. 6b), suggesting the stability of the two protein variants is differentially regulated even if they would complex with one another. We calculated the protein decay rate for each isoform. FLAG-TRIM46S had a shorter half-life of 1.9 h versus FLAG-TRIM46L of 4.3 h (Fig. 3d). Overall, these results showed the TRIM46S protein isoform is less stable than the TRIM46L protein.

Next, we tested whether TRIM46L and TRIM46S were subject to differential regulation by well-known protein degradation pathways. Over 90% of all proteins in cells are removed by the ubiquitin-proteasomal system (UPS) and the lysosomal pathway[68–72]. To investigate if the UPS pathway regulates protein degradation of TRIM46 proteins, we treated N2a cells expressing

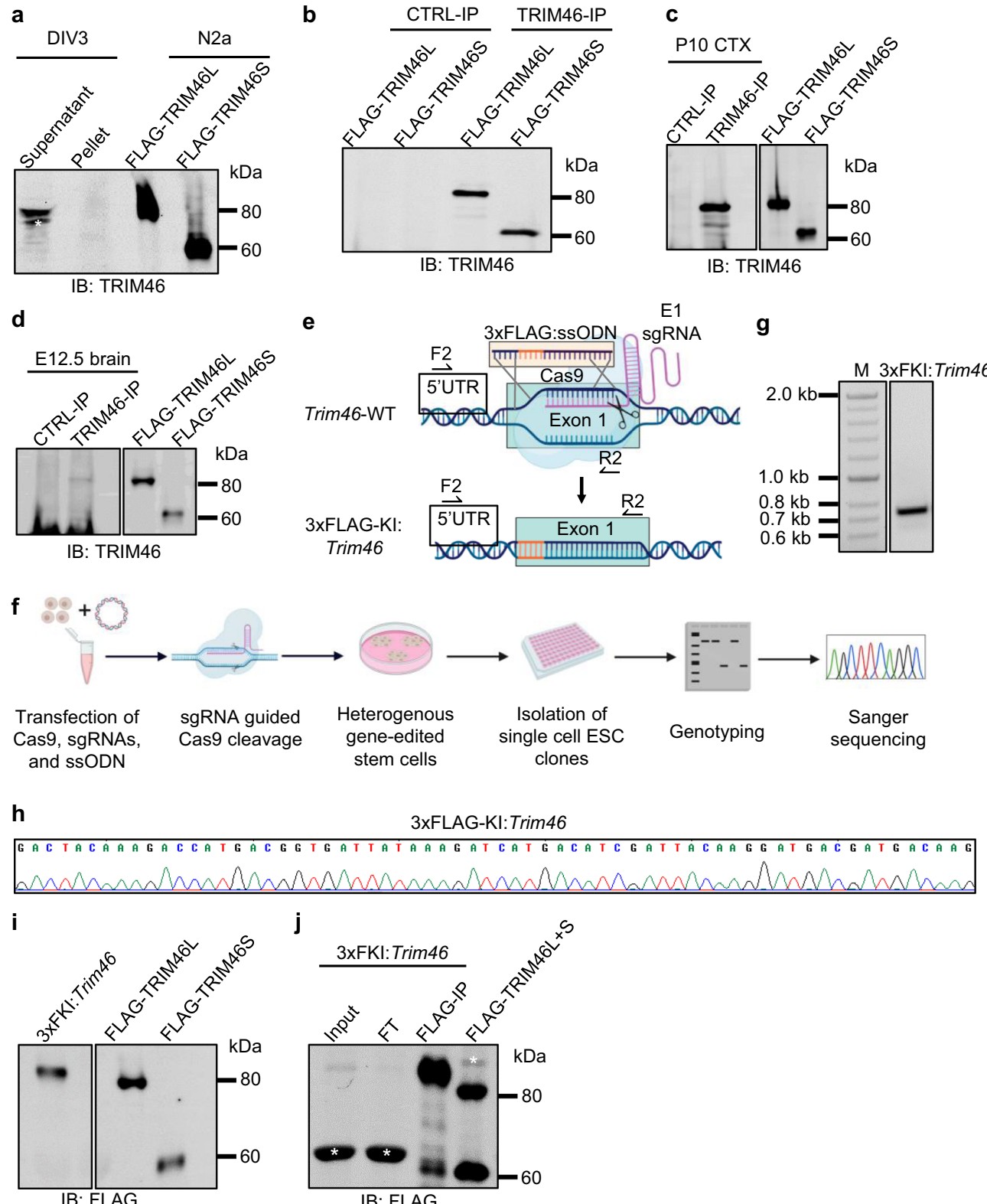

FLAG-TRIM46L or FLAG-TRIM46S with proteasome inhibitor MG132. We found FLAG-TRIM46S was stabilized by MG132 in a dose-dependent manner (Supplementary Fig. 6c). We chose 10 µM MG132 for subsequent experiments because higher concentrations appeared to induce cellular toxicity and because proteasome inhibition was evident with increased ubiquitinated proteins (Supplementary Fig. 7a). In comparison to DMSO control, 10 µM MG132 treatment upregulated both FLAG-TRIM46L and FLAG-TRIM46S proteins. However, the FLAG-TRIM46S protein was more sensitive to MG132 than the FLAG-TRIM46L protein (Figs. 3e and 3f).

We attempted the MG132 experiment on DIV 1 neurons derived from 3xFKI:*Trim46* ESCs. However, in 3xFLAG-KI neurons treated with MG132 from 10 nM to 100 µM there was

**Fig. 2 Presumptive endogenous TRIM46S protein is undetectable. a** Immunoblot for TRIM46 protein in the supernatant and pellet fraction from total lysate of WT ESC-derived neurons on DIV 3. Whole cell lysates from N2a cells expressing FLAG-TRIM46L and FLAG-TRIM46S proteins were used as a positive control. Asterisk (*) denotes nonspecific band. **b** Immunoprecipitation (IP) using protein A magnetic beads incubated with TRIM46 antibody or CTRL IgG to pulldown FLAG-TRIM46L and FLAG-TRIM46S proteins expressed in N2a cells. **c** P10 cortices (CTX) and **d** E12.5 mouse brains used in TRIM46-IP experiment to detect endogenous TRIM46 protein isoforms. CTRL-IP used IgG for immunoprecipitation. The two lanes on the right are positive controls showing the position of the expected protein isoforms ran on the same gel: N2a cells expressing FLAG-TRIM46L and FLAG-TRIM46S. **e** CRISPR/Cas9 strategy to generate 3xFLAG-knock-in:*Trim46* (3xFKI:*Trim46*) cell line from WT ESC using single-stranded oligo DNA (ssODN) along with exon 1-sgRNA (E1 sgRNA) targeting exon 1. After homology-dependent repair, 3xFLAG is introduced to the endogenous *Trim46* locus. F2 and R2 indicate positions of genotyping primers annealing to 5′UTR and exon 1 of *Trim46*. **f** Schematic CRISPR/Cas9-based process in producing gene edited cell lines through lipofectamine-based co-transfection of Cas9-sgRNA plasmid, ssODN, and a GFP reporter plasmid. CRISPR/Cas9-edited cell lines were generated by selecting GFP+ colony and diluting cells into 96-well plates for screening. Identification of single-cell derived colonies was performed by genotyping and verified by Sanger sequencing. **g** PCR results showing genomic DNA band representing the 3xFLAG knock-in element of ES cell line, 3xFKI:*Trim46* using primers shown in e. Expected size for band of knock-in is 741 bp. **h** Sanger sequencing confirming 3xFLAG DNA sequence in 3xFKI:*Trim46* cell line. **i** Western blot using FLAG antibody for endogenously tagged FLAG-TRIM46 protein from 3xFKI:*Trim46*. Expected protein sizes of 3xFLAG-TRIM46L and 3xFLAG-TRIM46S protein isoforms are 86 kDa and 63 kDa, respectively. The right two lanes show the positive control (but with 1XFLAG): FLAG-TRIM46L and FLAG-TRIM46S expressed in N2a cells. **j** FLAG-IP experiment to detect 3xFLAG-TRIM46 protein isoforms using FLAG magnetic beads against lysate from 3xFKI:*Trim46* ESC-derived neurons on DIV 3. The last lane is the positive control showing the positions FLAG-TRIM46L and FLAG-TRIM46S co-expressed in N2a cells. Asterisk (*) indicates various nonspecific bands. All experiments were repeated at least three times.

little change in ubiquitination levels (Supplementary Fig. 7a). We also extended the treatment time to 24 h, which did not upregulate the ubiquitin signals but did cause massive cell death at various concentrations (Supplementary Fig. 7b). The MG132 toxicity (probably proteasome-independent) in ESC-derived neurons precluded further investigation.

To test for differences in TRIM46 protein isoform degradation under the lysosomal pathway, we inhibited the pathway with NH4Cl in N2a cells expressing FLAG-TRIM46 proteins. We observed insignificant effects on TRIM46S expression by up to 10 mM NH4Cl but a substantial reduction after treatment of 50 or 100 mM NH4Cl due to cellular toxicity (Supplementary Fig. 6d). Importantly, TRIM46L and TRIM46S were not differentially affected and neither protein increased abundance (Fig. 3g), suggesting that TRIM46 protein is unlikely to be regulated by the lysosomal pathway. Overall, these results show TRIM46S proteins were less stable than TRIM46L proteins due to higher sensitivity to UPS-mediated protein degradation.

**Ablation of exon 10 results in loss of TRIM46 protein and reduced AnkG localization at the proximal axon.** To test the functional significance of exon 10 alternative splicing during axon formation, we generated an exon 10 deletion mutant using CRISPR/Cas9 and tested its effect on TRIM46 protein and AnkG localization to the proximal axon. We designed two sequence-specific guide RNAs targeting intronic sequences flanking exon 10. We then used the CRISPR/Cas9 system to knock out exon 10 (*Trim46*$^{\Delta E10/\Delta E10}$ or E10-KO) in 46C ES cells (Fig. 4a) so that the derived neurons would constitutively express only the exon-skipped E10− variant, allowing the study of loss of TRIM46L function during axon formation.

Targeted exon 10 deletion in WT ES cell line was verified by PCR genotyping (Fig. 4b). This E10-KO line contains two slightly different KO alleles as demonstrated by Sanger sequencing (Fig. 4c). We performed Western Blot analysis for endogenous TRIM46 protein using WT and E10-KO ESC-derived neurons on DIV 5 and confirmed the loss of 83 kDa TRIM46L proteins (Fig. 4d). The TRIM46S protein was not detected despite the loss of exon 10 (Fig. 4d), which is consistent with previous results showing lower stability of TRIM46S proteins (Fig. 3). We also tested pellets from E10-KO protein lysate and, again, confirmed the absence of TRIM46S protein (Supplementary Fig. 5b).

The loss of TRIM46 proteins was not due to transcriptional defects in E10-KO because there was no significant difference in total *Trim46* mRNA levels between control and mutant neurons

(Fig. 4e). We observed a general upregulation of total *Trim46* mRNA levels from EB D8 to DIV 7 in both WT and E10-KO neurons, indicating transcriptional activation. The lack of mRNA expression differences between control and mutant exon 10-KO neurons supports that transcriptional regulation alone is not responsible for the change in TRIM46 protein expression in the mutant neurons. Indeed, the lack of functional TRIM46 proteins in exon 10 KO demonstrates that exon 10 splicing regulation is an essential gatekeeper of TRIM46 protein function.

The major functional outcome of TRIM46 is to influence AnkG recruitment to the proximal axon (AIS)[4,22]. Therefore, we examined AnkG localization in E10-KO ESC-derived neurons. E10-KO and control parental WT ESCs were differentiated into neurons in parallel and stained with TRIM46 and AnkG (Fig. 4f). By DIV 7, WT ESC-derived neurons exhibited TRIM46 accumulation at the proximal axon as expected. By contrast, E10-KO neurons universally lacked the prominent TRIM46 staining in the corresponding region (Fig. 4f), consistent with loss of TRIM46 proteins revealed by Western immunoblots (Fig. 4d). Note that the TRIM46 antibody exhibited nonspecific somatic staining (Fig. 4f and Supplementary Fig. 1c).

Since no TRIM46 protein was detected in E10-KO neurons (Fig. 4d), exon 10 KO is essentially a *null* for TRIM46 function. We observed a significant decrease in the percentage of AnkG-positive neurons when comparing E10-KO (51%) vs. WT (86%) (Fig. 4g). Surprisingly, about half of E10-KO neurons retain the capacity for generating AnkG clusters at the AIS site. For those displaying AnkG staining at the proximal axon, we did not observe any significant difference in AnkG length or AnkG intensity between WT and E10-KO neurons (Figs. 4h and 4i). Therefore, TRIM46 is a significant regulator but not absolutely required for AnkG clustering at AIS (see more in Discussion). Together, these results showed that in the absence of exon 10 inclusion, the TRIM46 protein was not detectable and consequently AnkG was less effectively localized to the proximal axon.

We investigated whether neuronal polarity was impaired in E10-KO neurons, which could contribute to the AnkG phenotype. WT and E10-KO neurons were co-stained with axonal marker Tau1 and dendritic marker MAP2 at DIV 5 (Supplementary Fig. 8a). We then quantified the average dendritic intensity ($I_d$) and average axonal intensity ($I_a$) of Tau1. We calculated the Tau1 axonal polarity index $P = \frac{I_a - I_d}{I_a + I_d}$. $P > 0$ means polarized axonal distribution; $P = 0$ means uniform distribution; $P < 0$ means polarized dendritic distribution. As shown in

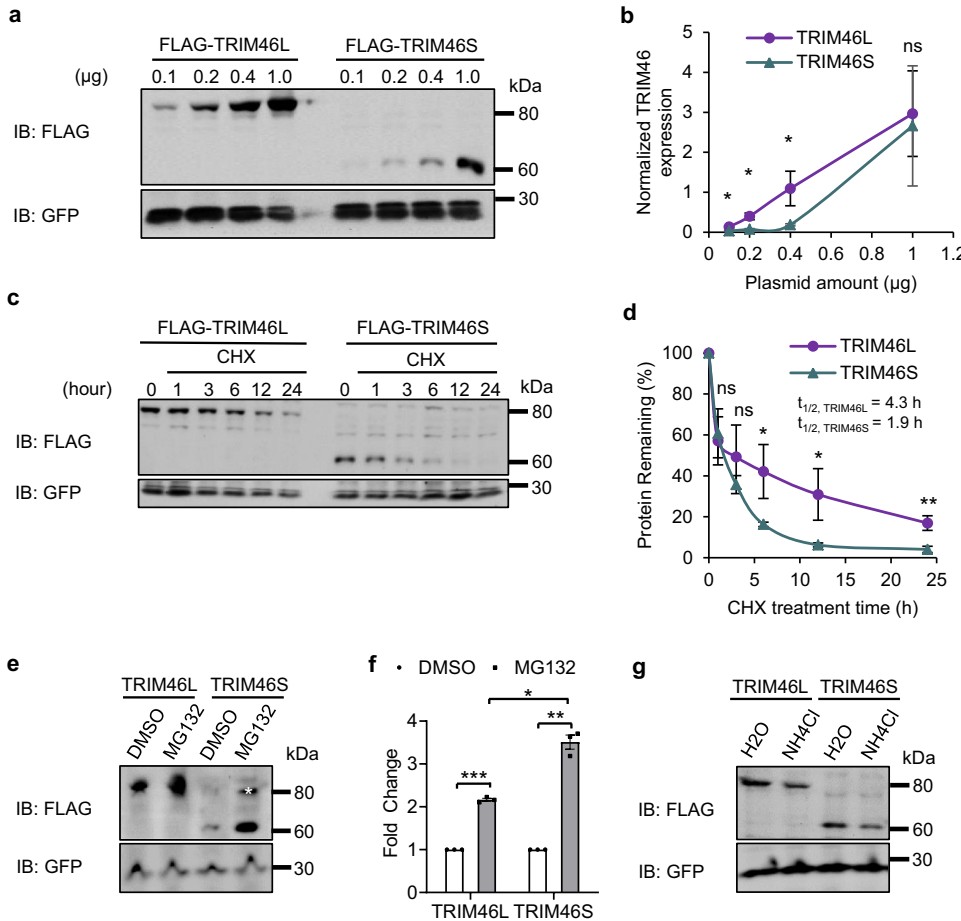

**Fig. 3 TRIM46S protein variant is less stable than TRIM46L protein. a** Expression of FLAG-TRIM46 proteins in N2a cells with transfection of increasing FLAG-TRIM46L/S plasmid amounts from 0.1 to 1 µg, compensated with CAGIG vector to make total transfection amount 1 µg. GFP is used as an internal control. **b** Quantification shows FLAG-TRIM46S protein is expressed substantially less than FLAG-TRIM46L protein with transfection of 0.1, 0.2, and 0.4 µg plasmids. Data is represented as mean ± SD. Statistics: *t*-test, one-tailed, paired; Statistically not significant, (ns); n = 3 independent experiments; (0.1 µg, \*p = 0.02331; 0.2 µg, \*p = 0.01438; 0.4 µg, \*p = 0.03036; 1.0 µg, ns = 0.25465). **c** Protein stability test in N2a cells transfected with 0.2 µg plasmids FLAG-*Trim46L* or FLAG-*Trim46S*. Lysates were collected at 0 h as the control versus cycloheximide (CHX) treatment over 1, 3, 6, 12, and 24 h. **d** The graph shows significantly less FLAG-TRIM46S protein remains over time in comparison to FLAG-TRIM46L protein after CHX treatment. Data is represented as mean ± SEM. Statistics: *t*-test, one-tailed, paired; n = 3 independent experiments; Statistically not significant (ns); (1 h, ns = 0.2928; 3 h, ns = 0.1077; 6 h, \*p = 0.03152; 12 h, \*p = 0.04761; 24 h, \*\*p = 0.007146). The CHX-based stability assay determined the half-life of FLAG-TRIM46S protein, which was significantly shorter than and FLAG-TRIM46L protein. **e** FLAG-TRIM46 protein isoforms expressing in N2a cells are substantially upregulated under proteasomal inhibition via MG132 treatment. Asterisk (\*) represents nonspecific bands. **f** The graph shows FLAG-TRIM46S protein is more sensitive to proteasomal repression than the FLAG-TRIM46L isoform. TRIM46 protein quantification between DMSO and MG132 is normalized to isoform of interest. Data is represented as mean ± SEM. Statistics: *t*-test, two-tailed, paired; n = 3 independent experiments; (FLAG-TRIM46L: DMSO v. MG132, \*\*\*p = 0.0009153; FLAG-TRIM46S: DMSO v. MG132, \*\*p = 0.004244; MG132: FLAG-TRIM46L v. FLAG-TRIM46S, \*p = 0.01427). **g** FLAG-TRIM46 protein expression is not affected by the lysosomal pathway as demonstrated by western blot for FLAG and GFP between H2O and NH4Cl treated N2a cells; n = 2 independent experiments showing the same results.

Supplementary Figures 8b, control neurons show $P_{Tau1} = 0.767$ while exon 10 knockout neurons exhibit $P_{Tau1} = 0.765$. Furthermore, most neurons possess a single axon in both control (86.4%) and exon 10 KO neurons (85.9%). There is essentially no difference among WT and mutants regarding the percentage of neurons with 2 axons (7.8% and 8.6% for control and exon 10 KO, respectively, $p = 0.83$, two-tailed, unpaired *t*-test) or without axons (5.8%, 5.5% for control and exon 10 KO respectively, $p = 0.87$, two-tailed unpaired *t*-test). Since Tau1 polarity and axon number are unaffected, the exon 10 knockouts do not appear to alter the initial step of axonal polarization but the subsequent formation of AnkG cluster. Therefore, temporal

control of *Trim46* splicing and expression regulates certain but not all aspects of axon formation.

We also investigated whether exon 10 knockout affected neuronal subtype identity to cause the AnkG phenotype. We stained WT and E10-KO neurons with glutamatergic marker (vGLUT1) and GABAergic marker (GAD67). We found no differences in population composition between WT and E10-KO ESC-neurons: the overwhelming majority are 97% glutamatergic neurons for both (Supplementary Fig. 8d and 8e). Given similar percentage of vGLUT1+ and GAD67+ cells between control and exon 10-KO neurons this shows the LOF phenotype is not due to alteration in neuronal populations.

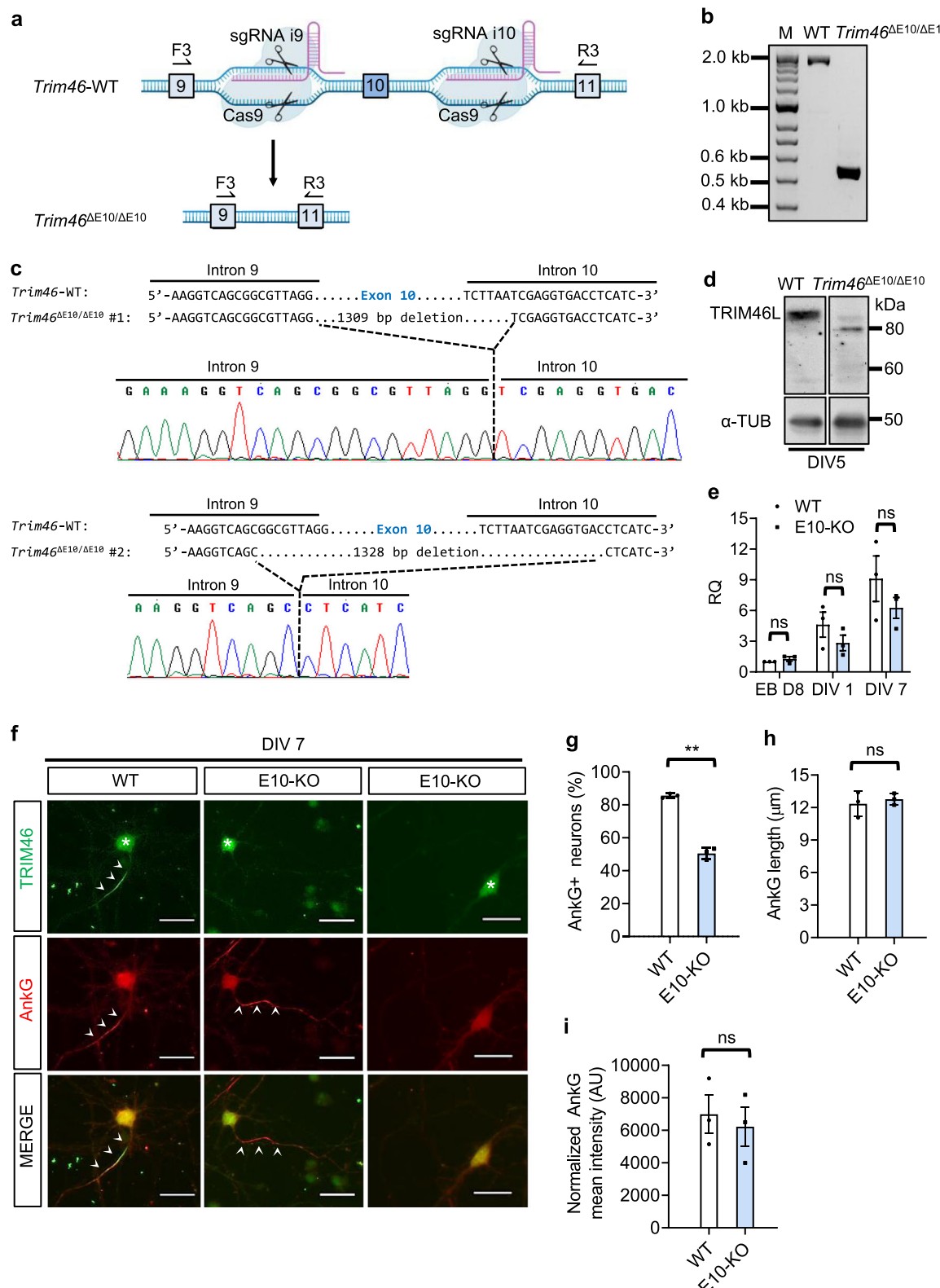

**Exon 8 inclusion leads to nonsense-mediated mRNA decay of *Trim46* transcripts and is developmentally downregulated**. We next studied another alternative exon (exon 8) of *Trim46*. The TRIM46 protein is translated from the exon 8 exclusion (E8−) isoform with a stop codon in exon 11 (Fig. 5a). Exon 8 inclusion results in frameshift and termination of the reading frame at a premature stop codon (PTC) in exon 9, making the inclusion

isoform a possible target of NMD for degradation without productive translation (Fig. 5a). We observe the inclusion of exon 8 declined dramatically from EB day 8 to DIV 1 and showed continuous downregulation in subsequent stages until DIV 7 (Fig. 5b). The ENCODE data show exon 8 inclusion decreases from E11 to E18 in developing cortices (Supplementary Fig. 2c). Therefore, exon 8 is developmentally regulated during axon formation.

**Fig. 4 Ablation of exon 10 results in loss of endogenous TRIM46L protein and leads to reduced AnkG assembly at the proximal axon. a** Schematic of CRISPR/Cas9 strategy for the deletion of *Trim46* exon 10 in WT ESC. SgRNAs i9 and i10 denote guide RNAs targeting introns 9 and 10 for exon 10 deletion through Cas9 activity. F3 and R3 indicate genotyping primers annealing to exon 9 and 11. Arrow represents edited locus showing change from WT to mutant $Trim46^{\Delta E10/\Delta E10}$ (E10-KO). **b** Genotyping results for WT and $Trim46^{\Delta E10/\Delta E10}$ cell line. Band sizes are 1863 bp, 530 bp, and 513 bp for WT, $Trim46^{\Delta E10/E10}$ allele 1, and $Trim46^{\Delta E10/\Delta E10}$ allele 2, respectively from 5 independent experiments. **c** Sanger sequencing of PCR product shows loss of exon 10 in genomic DNA from $Trim46^{\Delta E10/\Delta E10}$ cell line. Top and bottom panel shows exon 10 deletion in chromatograms of E10-KO alleles #1 and #2 aligned to WT sequence. **d** Western blot analysis from ESC-derived neurons on DIV 5 confirms loss of TRIM46L protein in E10-KO cell line from 6 independent experiments. Weak bands around 80 kDa are nonspecific. **e** RT-qPCR for total *Trim46* mRNA levels between WT and E10-KO from EB day 8, DIV 1, and DIV 7. Data is represented as mean ± SD; Statistics: *t*-test, two-tailed, unpaired: WT v. E10-KO; EB D8, ns = 0.28179786, DIV 1, ns = 0.28175346, DIV 7, ns = 0.3096017; WT, $n = 3$ biological replicates. **f** ESC-derived neurons on DIV 7 from WT and E10-KO stained for TRIM46 (green) and AnkG (red). Arrowhead denotes TRIM46 or AnkG signal at the AIS and asterisk (*) denotes nonspecific TRIM46 somatic staining. Scale bars, 10 μm. **g** Quantification between WT and E10-KO for AnkG-positive neurons (%), (**$p = 0.005941$); **h** AnkG length (μm), (ns = 0.3699); and **i** Normalized AnkG mean intensity in arbitrary units (AU), (ns = 0.6029); WT, $n = 160$ and E10-KO, $n = 216$ from 3 independent experiments. Data is represented as mean ± SEM. Statistics: *t*-test, two-tailed, paired; Statistically not significant, (ns).

To evaluate whether the exon 8 inclusion isoform is targeted by NMD, we treated cortical neurons with CHX to inhibit NMD. Compared to DMSO control, there was a significant (5.0-fold) increase in the E8+ variant in CHX-treated neurons (Figs. 5c and 5d, $p = 1.09 \times 10^{-10}$). To further confirm NMD regulation of *Trim46* transcripts, we also treated N2a cells with CHX. N2a showed a similar upregulation of the E8+ NMD isoform upon CHX treatment (Supplementary Fig. 9a and 9b).

To genetically confirm the E8+ isoform is regulated by NMD, we knocked out the essential NMD factor *Upf2* in the mouse cortex (*Upf2*-cKO) using the Emx1-Cre line[40,55]. We found that in *Upf2*-cKO cortices the E8+ isoform was substantially upregulated at E14.5 (8.7-fold, $p = 1.3 \times 10^{-5}$) and P1 (6.8-fold, $p = 1.5 \times 10^{-5}$, Figs. 5e and 5f). By comparison, exon 10+ transcripts relative to exon 10− transcripts at E15.5 were marginally changed (1.5-fold, Supplementary Fig. 9c and 9d), probably under indirect effects of NMD inhibition. A similar small effect was observed in CHX-treated DIV 1 primary cortical neurons (1.4-fold, Supplementary Fig. 9e and 9f), in contrast to the much larger fold change observed in exon 8+ transcripts (5.0-fold, Figs. 5c and 5d). Finally, we knocked down a core NMD effector UPF1 using verified *Upf1*-specific siRNA[43]. *Upf1* knockdown led to a modest but statistically significant increase in the exon 8 inclusion isoform probably owing to residual cellular UPF1 proteins (Figs. 5g and 5h). Taken together, our results demonstrate the *Trim46* E8+ isoforms are subject to NMD regulation.

We found the premature stop codon of the exon 8+ isoform is highly conserved across a multitude of species (including all annotated mammals), suggesting that NMD regulation of *Trim46* is under selection pressure and is biologically important (Fig. 5i). Given that NMD targets are translationally repressed, exon 8 splicing is a major mechanism restraining TRIM46 protein expression in neural progenitors and developing neurons[37–39]. This restraint is alleviated as exon 8 splicing decreases during development, resulting in increased TRIM46 protein expression. Consistent with this idea, the TRIM46 protein is up-regulated during differentiation of ESC-derived neurons and in developing neocortices (Figs. 1c, 5j, and Supplementary Fig. 1b). The above data, taken together, supports the conclusion that a decrease in exon 8 inclusion or a switch from the NMD-sensitive to the NMD-insensitive isoform of *Trim46* transcripts allows the increase in TRIM46 protein expression during axon formation.

**Ablation of *Trim46* exon 8 precociously induces TRIM46 proteins and enhances AnkG localization to the proximal axon.** To examine the functional significance of AS-NMD regulation on *Trim46*, we deleted exon 8 and its flanking introns from the genome to analyze the effects on TRIM46 expression and

subsequent AnkG localization. We designed sequence-specific guide RNAs targeting intronic sequences flanking exon 8. We used CRISPR/Cas9 to knock out exon 8 ($Trim46^{\Delta E8/\Delta E8}$ or E8-KO) in ES cells so that the derived neurons would constitutively express only the E8-skipped isoform, abolishing developmental use of the E8+ variant and NMD regulation (Fig. 6a). Targeted exon 8 deletion in ES cells was verified by PCR genotyping (Fig. 6b) and Sanger sequencing (Fig. 6c).

To investigate the effect of exon 8 KO on TRIM46 expression, we collected protein lysates from WT and E8-KO ESC-derived neurons on DIV 3 and DIV 5. Western blot analysis showed TRIM46L proteins in E8-KO were substantially upregulated at both stages (Figs. 6d–6g). This is consistent with the prediction that exon 8 deletion de-represses TRIM46 protein expression. Despite upregulation of TRIM46L proteins, the TRIM46S protein isoform was again undetectable.

At DIV 1 and DIV 7 there was no obvious difference in total *Trim46* mRNA levels between control and E8-KO mutant neurons (Fig. 6h). The general lack of mRNA expression differences between control and exon 8-KO neurons supports that transcriptional regulation alone is not responsible for the change in TRIM46 protein expression in mutant neurons. However, we note on EB D8 there was significant mRNA upregulation in E8-KO neurons compared to control. This is because in WT neurons exon 8 is largely included at EB day 8 leading to NMD of *Trim46* transcripts. Exon 8 deletion prevents NMD of *Trim46* transcripts, effectively increasing its steady-state abundance. Therefore, this data confirms the significant role of exon 8 splicing and NMD in regulating *Trim46* transcript abundance.

To understand the functional consequence of precocious TRIM46 induction by exon 8 deletion, we stained WT and E8-KO ESC-derived neurons on DIV 3 for TRIM46 and AnkG (Fig. 6i). We found that the TRIM46 length at the proximal axon in E8-KO neurons was significantly longer than in WT (14.7 μm vs. 9.1 μm, Fig. 6j). In addition, the normalized mean TRIM46 intensity was significantly higher in E8-KO than in WT neurons, showing more TRIM46 proteins clustered at the proximal axon (Fig. 6k). This also indicates that the TRIM46 amount localized to the future axon is correlated with total TRIM46 expression.

About 75% of WT neurons specified their axons and displayed AnkG clustering at the proximal axons with an average length of 9.6 μm (Figs. 6l and 6m). We observed a significantly higher proportion of E8-KO neurons (93%) containing AnkG clusters at the proximal axon (Fig. 6l). AnkG length in E8-KO neurons was substantially longer than WT neurons (Fig. 6m). Interestingly, there was no change in the normalized mean AnkG intensity between E8-KO and WT among AnkG+ neurons (Fig. 6n). As a whole, these observations support the conclusion that deletion of

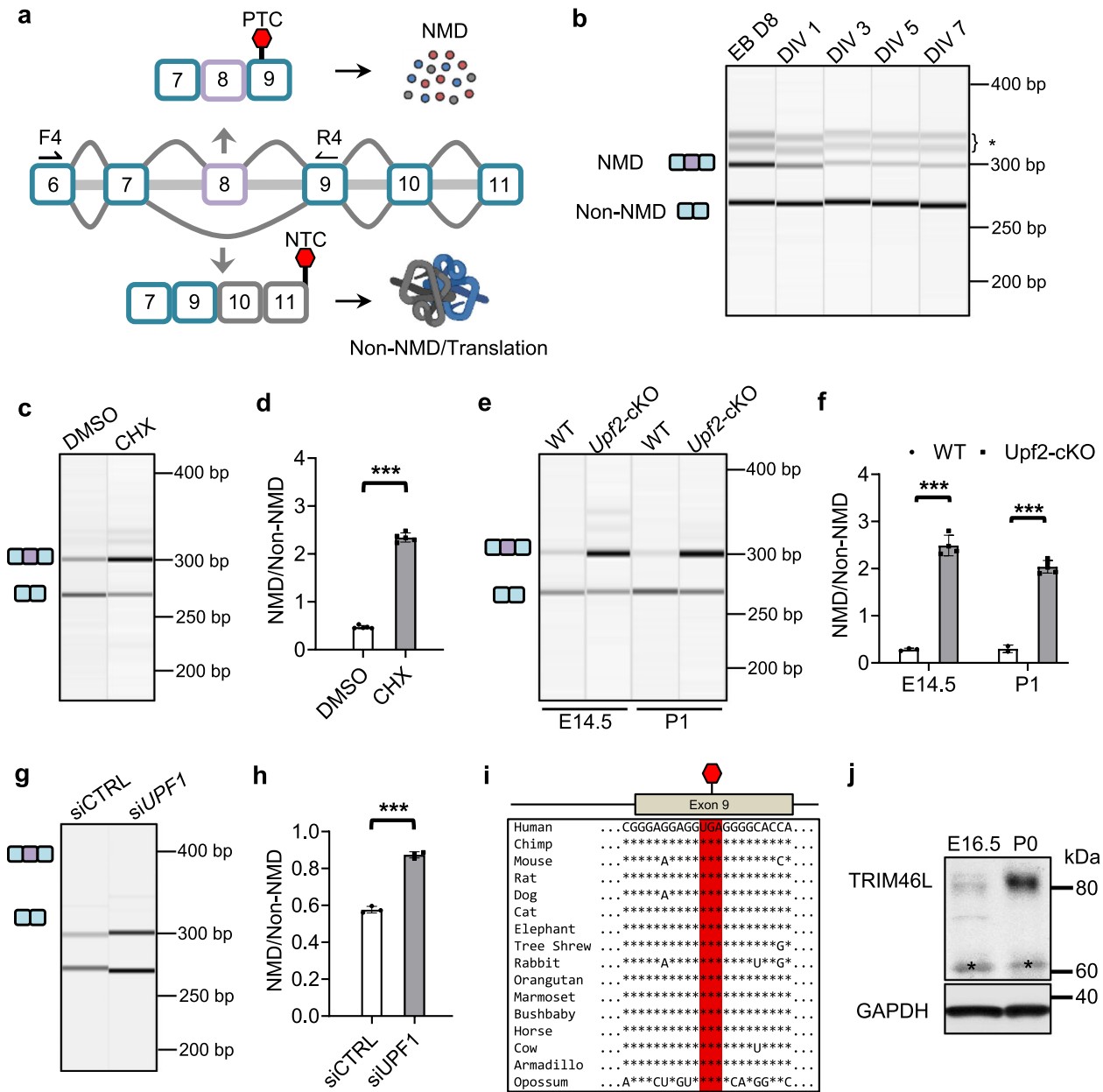

**Fig. 5 Exon 8 inclusion is developmentally downregulated and leads to nonsense mediated decay of *Trim46* transcripts. a** Schematic showing alternative splicing regulation on *Trim46* expression through AS-NMD control of exon 8. The inclusion of exon 8 alters the reading frame and leads to a premature termination codon (PTC) in exon 9 while exon 8 exclusion leads to normal termination codon (NTC) in exon 11. Red stop signs represent stop codons. F4 and R4 indicate the location of primers for RT-PCR analysis of *Trim46* exon 8 splicing detecting NMD isoform (E8+) and non-NMD isoform (E8-). **b** Developmental *Trim46* mRNA expression of NMD and non-NMD isoforms in WT ESC-derived neural progenitors and neurons from multiple stages during axon formation: day 8, DIV 1, 3, 5, and 7. $N = 2$ shows the same results. Bracketed asterisk points to nonspecific/heteroduplex bands. **c** Gel image highlighting substantial increase of E8+ *Trim46* isoform under NMD inhibition (with CHX) from DIV 1 primary cortical neurons cultured from WT E13.5 neocortices from 2 independent experiments. **d** Normalization of *Trim46*-NMD transcripts by the non-NMD transcripts in DMSO and CHX-treated cortical neurons ($n = 5$). Data is represented as mean ± SD; Statistics; $t$-test, two-tailed, unpaired; (***$p = 1.0892 \times 10^{-10}$). **e** Gel image shows upregulation of *Trim46*-NMD isoform in the absence of UPF2 in mouse neocortices at E14.5 and P1 from 3 independent experiments. **f** Normalization of *Trim46*-NMD isoform by non-NMD isoform in WT and *Upf2*-cKO neocortices. Data is represented as mean ± SD. Statistics; $t$-test, two-tail, unpaired; (E14.5: WT, $n = 3$ and KO, $n = 4$, ***$p = 1.2823 \times 10^{-5}$); (P1: WT, $n = 2$ and KO, $n = 5$, ***$p = 1.45171 \times 10^{-5}$). **g** Depletion of *Upf1* displays an increase in E8+ *Trim46* isoform from 2 independent experiments. **h** Quantification of *Trim46*-NMD transcripts in siCTRL and si*UPF1* transfected N2a ($n = 3$). Data is represented as mean ± SD. Statistics: t-test, two-tailed, unpaired; (***$p = 2.92045 \times 10^{-5}$). **i** The PTC generated by the E8+ isoform of *Trim46* is conserved in annotated species. UGA represents PTC highlighted in green. **j** Western blot shows upregulated TRIM46 protein expression during development in E16.5 and P0 neocortices. Asterisk (*) denotes nonspecific bands around 60 kDa.

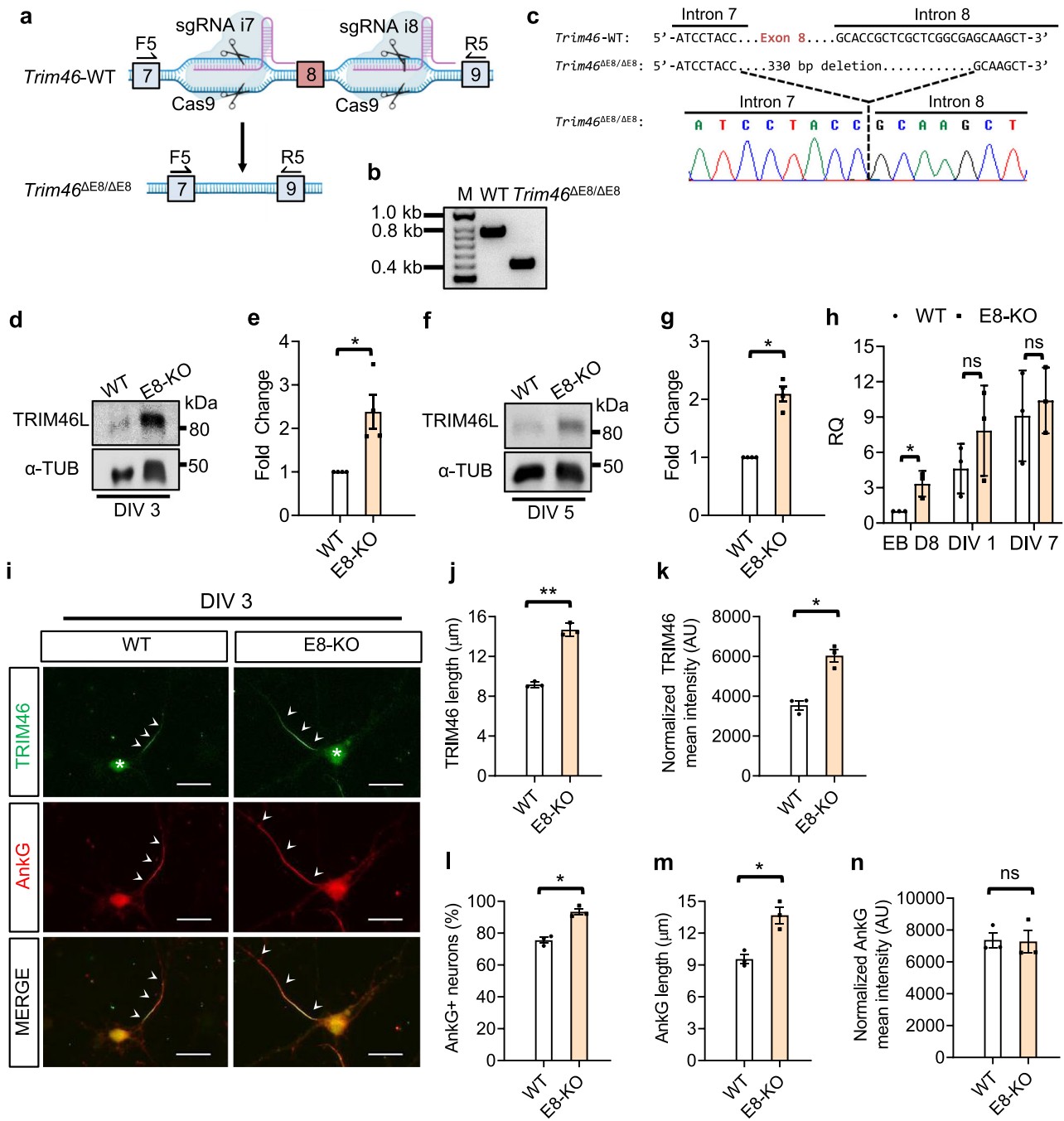

NMD-associated exon 8 increases accumulation of TRIM46 proteins and enhances AnkG localization to the AIS during axon formation. The correlation between TRIM46 and AnkG lengths further suggests TRIM46 plays an instructive role for AnkG localization.

We further analyzed whether neuronal cell types or neuronal polarity are affected in E8-KO. We found loss of exon 8 influenced neither Tau1 distribution nor polarization of neurons (Supplementary Fig. 10a and 10b). Axon numbers in E8-KO neurons were likewise unaffected (Supplementary Fig. 10c). We found no differences in population composition between mutant E8-KO ESC derived-neurons (97% glutamatergic and 3% GABAergic, Supplementary Fig. 10d and 10e). Together, the data show neuronal polarity and identity do not contribute to the AnkG phenotype in E8-KO neurons.

**Relationship between exon 8 and exon 10 splicing.** Since temporal splicing of exons 8 and 10 is concurrent during axon formation, we investigated whether regulation of the two exons was interconnected. If splicing of exons 8 and 10 are independent, neurons can express four *Trim46* isoforms (E8+E10−, E8+E10+, E8−E10−, E8−E10+). If one exon splicing fully determines the other, neurons would express only two of these four isoforms (E8+E10− and E8−E10+), since exon 8 inclusion is correlated with exon 10 skipping (Figs. 1f and 5b). To distinguish these two possibilities, we designed primers to amplify regions from exon 7 to exon 11. As shown in Supplementary Fig. 11a, we observed all four isoforms in all developmental stages.

More importantly, the percentage of exon 10 inclusion in exon 8-containg mRNAs (E8+E10+/E8+) increased dramatically

**Fig. 6 Ablation of exon 8 increases TRIM46 protein and leads to enhanced AnkG localization to the proximal axon. a** Schematic CRISPR/Cas9 strategy to convert parental WT ESC to mutant exon 8-KO. Graphical representation of *Trim46* locus showing sgRNAs targeting introns 7 and 8 to generate exon 8 deletion. F5 and R5 denote approximate location of genotyping primers to detect removal of exon 8. Arrow represents edited locus showing change from WT to mutant *Trim46*$^{\Delta E8/\Delta E8}$ or (E8-KO). **b** Genotyping results of WT and *Trim46*$^{\Delta E8/\Delta E8}$ ESC. Band sizes are 739 bp and 409 for WT and *Trim46*$^{\Delta E8/\Delta E8}$, respectively from 5 independent experiments. **c** Sanger sequencing of PCR product from genomic DNA confirms exon 8 deletion in E8-KO ES cell line. Top panel shows sequence alignment between WT and E8-KO lines and the bottom panel shows chromatogram of E8-KO mutant highlighting sequence of intron 7 preceding the 330 bp deletion of exon 8 and the immediate downstream sequences. **d–g**. Western blot highlights upregulated TRIM46 protein expression in exon 8 deletion mutants from ESC-derived neurons on DIV 3 and DIV 5. Protein quantification for TRIM46L protein between WT and E8-KO neurons on DIV 3 and DIV 5; *n* = 4 independent experiments; Data is represented as mean ± SEM. Statistics: *t*-test, two-tailed, paired; (DIV 3: WT v. E8-KO, *$p$ = 0.02447; DIV 5: WT v. E8-KO, *$p$ = 0.02008). **h** RT-qPCR for *Trim46* total mRNA levels between WT and E8-KO from EB day 8, DIV 1, and DIV 7. Data is represented as mean ± SD; Statistics: *t*-test, two-tailed, unpaired: WT v. E8-KO; EB D8, ns = 0.02091317, DIV 1, ns = 0.27058758, DIV 7, ns = 0.65857523; WT, *n* = 3 biological replicates. **i** ESC-derived neurons of WT and E8-KO on DIV 3 stained for TRIM46 (green) and AnkG (red). Arrowhead denotes TRIM46 signal at the proximal axon and asterisk (*) denotes nonspecific TRIM46 somatic staining. Scale bars, 10 μm. **j, k**. Quantification between WT and E8-KO neurons on DIV 3 for TRIM46 length (μm), (**$p$ = 0.001351) and normalized TRIM46 mean intensity in arbitrary units (AU), (*$p$ = 0.04061). **l** The deletion effect of *Trim46* exon 8 between WT and E8-KO neurons on DIV 3 for percentage of AnkG-positive neurons at the AIS, *$p$ = 0.02048; **m**, AnkG length (μm), *$p$ = 0.01743; and **n** normalized AnkG mean intensity in arbitrary units (AU), ns = 0.7097. Data is represented as mean ± SEM; WT, *n* = 130 and E8-KO, *n* = 154 which represent 3 independent differentiation experiments; Statistics: *t*-test, two tailed, paired.

during axon formation rather than staying low (8.1%, 36%, 56%, 65% at EB D8, DIV 1, 3, 5, and 7, respectively, Supplementary Fig. 11b). That is, exon 8-containing mRNA did not necessarily tend to skip exon 10. Instead, it contained or lacked exon 10 depending on the differentiation status. Exon 8-containing mRNA tended to lack exon 10 at EB D8 but gradually increased exon 10 inclusion till DIV 7, just like exon 8-skipping mRNA (Supplementary Fig. 11b).

To directly test causality, we asked whether exon 8 knockout affects exon 10 splicing, and vice versa. We found exon 8 inclusion did not change in exon 10-KO neurons in comparison to control ES-derived neurons during differentiation (Supplementary Fig. 11c and 11d). Similarly, exon 10 inclusion was not significantly affected in E8-KO neurons throughout neuronal differentiation from EB stage to maturing neurons at DIV 7 (Supplementary Fig. 11e and 11f). We also note that in PTBP2-cKO brains, exon 10 splicing changes (Figs. 1g-1i), but exon 8 splicing does not (Supplementary Fig. 12a and 12b), showing that exon 10 splicing does not affect exon 8 splicing. Therefore, exon 10 splicing itself is not a determinant of exon 8 splicing, and vice versa. Our data suggest that exons 8 and 10 splicing are likely coordinated by the differentiation signal upstream of their *trans*-regulators. The signal (either intrinsic or extrinsic or both) influences two different (possibly overlapping) sets of *trans* regulators that control exon 8 and exon 10 splicing.

**Exon 8 and Exon 10 splicing enforces tissue-specific TRIM46 expression.** Since TRIM46 is a significant determinant of axon formation and only neuronal tissues can generate axons, TRIM46 expression may be subject to stringent tissue-dependent regulations. We therefore examined RNA-seq data of adult mouse tissues generated by the ENCODE project. The steady-state *Trim46* mRNA levels are substantially higher in the brain tissues than non-neural tissues (mean TPM: 68.5 vs 8.0; Fig. 7a). Notably, this difference does not solely reflect differential transcriptional regulation, given NMD targeting of *Trim46* exon 8+ isoforms. The exon 8+ isoform was roughly 7.7-fold more efficiently degraded than the exon 8− isoform based on *Upf2*-cKO analyses (Figs. 5e and 5f). The more inclusion of exon 8, the more underestimation of *Trim46* transcription activity using steady-state mRNA levels. Indeed, exon 8 is overwhelmingly included in non-neural tissues even at the steady-state level (mean PSI: 15; Fig. 7b). After excluding the NMD effect, the mean *Trim46* TPM levels were estimated to be 288 in the brain tissues and 56 in the non-neural

tissues, indicating about a 5-fold difference in transcriptional outputs.

The steady-state *Trim46* mRNA levels do not precisely forecast the expression output of functional TRIM46 proteins. Under AS-NMD control, *Trim46* exon 8+ isoforms are degraded without productive translation; only the exon 8 skipping isoform is translationally productive. The expression difference between brain and non-brain tissues of the exon 8− isoform is substantially larger (mean TPM: 35.6 vs 0.9, or about 40-fold; Fig. 7c) than that of the total *Trim46* mRNA level. Besides exon 8, splicing of exon 10 also influences the final output of functional TRIM46 proteins, assuming TRIM46S remains unstable in non-neural tissues. Indeed, exon 10 is differentially spliced between neural and non-neural tissues (mean PSI: 96 vs 27, Fig. 7d). TRIM46L expression measured by the exon 10+ isoform is substantially larger in the brain than in non-neural tissues (mean TPM: 66 vs 1.5, or 44-fold; Fig. 7e).

Therefore, while estimated *Trim46* transcription exhibits on average 5-fold difference (up to 8-fold by steady-state mRNA levels) between the brain and non-neural tissues, exon 8 and exon 10 splicing amplifies the difference of effective expression outputs to 40- and 44-fold, respectively. To further quantify this divergent expression, we calculated index *τ*, a robust metric of tissue specificity, where 1 means expression in a single tissue, and 0 means equal expression among all tissues[73]. The *τ* values for the estimated transcriptional output, steady-state mRNA level, exon 10+ isoform, and exon 8− isoform are 0.38, 0.56, 0.80, and 0.84, respectively (Fig. 7f). In summary, exon 8 and exon 10 splicing, as well as NMD control, are essential parts of the regulatory mechanisms enforcing brain-specific expression of TRIM46.

## Discussion

Here, we report the gene regulatory mechanisms controlling temporal and tissue-specific expression of TRIM46, one of the earliest proteins localized to prospective axons and postulated to be a determinant of axonal fate. We found the TRIM46 protein is undetectable prior to axon formation and is induced at the onset of axonogenesis. Surprisingly, *Trim46* mRNA is well transcribed before axon formation. The developmental up-regulation of the TRIM46 protein is orchestrated by alternative splicing of exons 8 and 10. Exon 8 splicing determines mRNA stability of *Trim46* transcripts, whereas exon 10 splicing influences protein stability of TRIM46 protein variants.

Before axon formation, *Trim46* pre-mRNA is alternatively spliced to generate isoforms that restrain the expression of

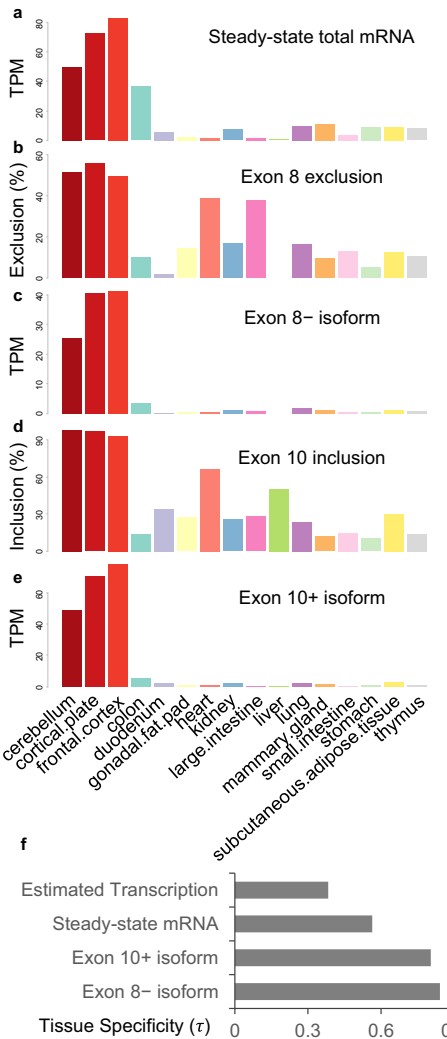

**Fig. 7 Exon 8 and Exon 10 splicing enforces tissue-specific expression of TRIM46. a** Steady-state total *Trim46* mRNA expression (TPM) of mouse adult tissues. RNA-seq data was obtained from ENCODE (Thomas Gingera's lab). The neural tissues are highlighted in red. The same dataset was used to calculate **b** exon 8 exclusion (%), **c** exon 8− isoform expression level (TPM), **d** exon 10 inclusion (%), **e** exon 10+ isoform expression level (TPM). **f** tissue specificity index $\tau$ of estimated transcriptional output, steady-state mRNA levels, exon 10+ isoform, and exon 8− isoform.

functional TRIM46 protein. Around the time when NPCs differentiate into post-mitotic neurons, these restraints are removed to allow the TRIM46 protein to accumulate due to splicing regulation of exons 8 and 10. We found exon 10 is repressed by PTBP2 during early neuronal differentiation but exon 8 is not, meaning they are separately regulated. Therefore, a multi-layered control, encompassing two independently-regulated alternative splicing events, NMD, and protein stability regulation coordinates expression of TRIM46 proteins to promote axon specification.

To what extent and which AS-NMD controls have biological function remains mostly unknown. While AS-NMD regulation appears pervasive and has gained appreciation for its role in finetuning gene expression, genetic testing for the necessity of a specific AS-NMD control is rarely reported. As NMD isoforms are often minor isoforms, most AS-NMD controls only modestly affect gene output[74]. We generated exon 8 knockout cells and demonstrated the biological significance of exon 8 AS-NMD

regulation. As exon 8 does not participate in encoding a functional TRIM46 protein, the impact of its splicing on the TRIM46 protein level further showcases the functional significance of RNA processing and control elements outside of protein-coding sequences.

Alternative splicing coupled with differential stability control of protein isoforms represents another mechanism of adjusting gene output through alternative splicing. To our knowledge, no precedent other than *Trim46* has been reported for this mechanism. Our study comprehensively demonstrated that endogenous TRIM46S proteins encoded by the E10− isoform are unstable compared to the E10+ TRIM46L proteins. First, two authenticated TRIM46 antibodies detected no endogenous TRIM46S proteins in total lysates or insoluble fractions of WT and E10-KO neurons. If TRIM46S and TRIM46L had similar stability, E10-KO neurons would express TRIM46S at the same level as WT neurons express TRIM46L. Second, enrichment through immunoprecipitation was still unable to detect TRIM46S proteins. Third, endogenous FLAG-tagged TRIM46 proteins were detected by the anti-FLAG antibody for the long isoform but not the short isoform. Fourth, TRIM46S protein had a substantially lower half-life than TRIM46L. Fifth, TRIM46S was more up-regulated than TRIM46L upon inhibition of the proteasome degradation pathway. Therefore, the regulation of exon 10 splicing mainly controls the total output of TRIM46 expression rather than generating proteins of varied functions.

*Trim46* would be the first example where two alternative exons coupled with two different downstream mechanisms converge to influence the total gene output. This integrated regulation is effective not only during axon formation but also for neural-specific expression of TRIM46 proteins. The difference in expression output between neural and non-neural tissues is substantially augmented by splicing of either exon (Fig. 7) and may be further enhanced by compounding their splicing effects. Absolute exon 8 inclusion or exon 10 exclusion can result in natural "knockout" of the *Trim46* gene product. However, alternative splicing outcomes are rarely all or none (i.e., PSI equal to 100 or 0), as seen for exon 8 and exon 10. They appear as double safeguards constituting a stringent mechanism to assure neural-specific expression of TRIM46. The ramification of TRIM46 expression in non-neural tissues is unclear but may relate to unique functions of TRIM46 for axonogenesis.

The exon 8 and exon 10 knockout alleles shed new light on TRIM46's role in axon specification. Exon 8 knockout precociously induces TRIM46 expression and consequently TRIM46 localization to future AIS sites. The increase in TRIM46 length at the AIS site lengthens the AnkG clustering along the axon, suggesting TRIM46 plays an instructive role for AnkG localization and AIS formation. Interestingly, the increase in the mean TRIM46 intensity at the AIS site did not change the mean AnkG intensity, indicating that TRIM46 functions to qualitatively mark the AIS location (and boundary) during the seeding step instead of to quantitatively determine the AnkG amount to be recruited to the site.

Exon 10 knockout is essentially *Trim46* null, expressing little functional TRIM46 proteins. Consistent with the instructive role of TRIM46, the percentage of AnkG+ neurons dropped substantially in the absence of TRIM46. However, about half of neurons still retained AnkG+ at the AIS with normal AnkG length and intensity. Therefore, TRIM46 is not always required for AnkG localization or clustering. This insight implies the need to search for additional TRIM46-like proteins that could compensate for the loss of TRIM46 proteins. The result is also consistent with the notion that TRIM46 and the yet-to-be-identified TRIM46-like proteins largely function to seed the AIS site rather than stoichiometrically determine AnkG density at the site. The

overall reduction of the seeding proteins reduces the probability of marking an AIS site. However, once a site is tagged for AIS, the amount of AnkG recruited to the site is independent of the level of seeding proteins at that site.

The exon 10 knockout phenotype appears less pronounced than previously reported *Trim46* knockdown neurons, which could be due to the timing and kinetics of TRIM46 deletion. van Beuningen et al.'s isolated hippocampal cultures and cortical cultures from E18 rat brains, and transfected the cultures with shRNA at DIV 1 and cortical cultures at DIV 4[4]. Because shRNA typically takes 2 or more days to deplete proteins (depending on shRNA efficiency and protein stability), it is likely TRIM46 is already well expressed and assumes functions of controlling AIS formation before being knocked down. In other words, van Beuningen's method acutely cuts off the supply of TRIM46 in the midst of exerting its cellular function. In E10-KO neurons, TRIM46 is not detectable at any time. Although this may sound worse than the knockdown, the absence may trigger compensatory mechanisms earlier, which provides neurons more time to adjust to the loss of TRIM46, e.g., by inducing the expression and accumulation of TRIM46-like molecules earlier. Given that TRIM46 is not absolutely required for AnkG accumulation, functionally redundant proteins likely exist to compensate in the absence of TRIM46. Second, RNAi-mediated knockdown is well known to have secondary effects including off-targets, which could worsen the phenotype or sensitize neurons to the depletion of TRIM46.

Accumulating evidence has demonstrated the pivotal role of alternative splicing control for mammalian axonogenesis[28]. Programming a large set of neural-specific alternative splicing events accompanies early axon formation[32]. Many of these are enriched in genes encoding cytoskeleton-associated proteins. These proteins presumably modify their enzymatic activities or their interactions with cytoskeletons through alternative splicing to regulate axon formation, as exemplified by *Shtn1* and *Cdc42* that control axonal growth[75,76]. Many AIS genes, including AnkG, are subject to RBFOX-mediated alternative splicing control during AIS formation[33]. *Trim46* exons were not reported to alter splicing in the *Rbfox* triple knockout. We found *Trim46* exon 10 is repressed by PTBP2. Notably, the PTBP2-mediated splicing program generally precedes the RBFOX-mediated splicing program[33]. Accordingly, TRIM46 expression and localization at AIS precedes the expression and localization of AnkG. These results strengthen the notion that developmental splicing programming is an integrated part of a larger gene expression regulatory network that coordinates axonogenesis. Future work should continue to elucidate the functional significance of these alternative splicing events for axon biology.

## Methods

**Cell culture, RNAi transfection, and CHX treatment**. Neuro-2a cells (mouse neuroblastoma, ATCC #CCL-131) were cultured in complete feeding media consisting of Dulbecco's Modified Eagle's Medium (DMEM) (Thermo Fisher), 10% fetal bovine serum (FBS) (Gibco), and 1x GlutaMAX (Gibco). We performed siRNA knockdown experiments using Lipofectamine RNAiMax (Thermo Fisher), Silencer Select siRNAs (Life Technologies, si*UPF1*, s72878), and Silencer Negative Control siRNA (Life Technologies, AM4615). Cells were plated overnight at 175,000 per well in 6-well plates at 2 ml, transfected according to manufacturer's instructions, and total RNA for RNAi experiments was collected forty-eight hours post-transfection. To evaluate NMD regulation, Cycloheximide (CHX) (Fisher Scientific, 50490338) was used at a final concentration of 0.5 mg/ml forty-eight hours after plating, and RNA was collected 8 h post-CHX treatment for downstream analysis.

**RNA extraction, cDNA synthesis, RT-qPCR, and QIAxcel quantitative capillary electrophoresis analysis**. Total RNA from mouse embryonic and postnatal neocortices or N2a samples were extracted using TRIzol reagent (Life Technologies, 15596–018) according to the manufacturer's instructions. Isolated RNA was treated with 4 units of Turbo DNase I (Ambion, AM2239) at 37 °C for 35 min to

degrade any remaining DNA before cDNA synthesis. After the DNase treatment, RNA was purified using phenol-chloroform (pH 4.5, 97064–744). RNA concentrations were measured using Nanodrop 2000c (Thermo Fisher). RNA conversion to first-strand cDNAs were synthesized in 20 μl reaction using 200 units of M-MLV reverse transcriptase, (Promega, M1705) 30 μM random hexamers, and a total of 1 μg RNA input. RT-qPCR were performed with Quantstudio 6 Real-Time PCR instrument in 10 μl reaction with 0.3 μl of 1:10 diluted cDNA, 5 μl of 2x SYBR Green PCR master mix (Life Tech), and 0.3 μM forward and reverse primers (Supplementary Data 1) followed by analysis via Quant Studio Real-Time PCR software. For QIAxcel analysis, 15 μl RT-PCR reactions were conducted using 6 μl of 1:10 diluted cDNA, 0.25 μM forward and reverse primers, and amplified via Taq DNA polymerase (New England Biolabs, M0267E) according to manufacturer's instructions. RT-PCR products were run through a digital capillary electrophoresis system using QIAxcel Advanced (QIAGEN), DNA Screening Kit (QIAGEN, 929004), Alignment Markers QX 15–600 bp (QIAGEN, 929530) and 15 bp/3 kb (QIAGEN, 929522), and Size Markers 25–500 bp (QIAGEN, 929560) and 100 bp-2.5 kb (QIAGEN, 929559) to separate and quantify bands. Visualization of RT-PCR products was performed using ScreenGel software (QIAGEN), and statistical analyses of calculated concentrations of target bands were carried out in Microsoft Excel.

**DNA transfection, MG132 or NH4Cl treatment, and CHX protein stability assay**. For DNA transfection experiments, we used GeneTran-III (BioMiga) along with plasmids pCAGIG-FLAG-*Trim46L*, pCAGIG-FLAG-*Trim46S*, pEGFP-C1-*Trim46L*, and pEGFP-C1-*Trim46S*. For protein expression analysis, we transfected the specified amount of *Trim46L* or *Trim46S* plasmid together with CAGIG and adjusted to a final plasmid amount totaling 1.0 μg: given 0.1 μg CAGIG-FLAG-*Trim46L*, we added 0.9 μg CAGIG vector plasmid totaling to 1.0 μg: (e.g., 0.2/0.8, 0.4/0.6, etc.). Forty-eight hours post transfection, cells were treated with DMSO or H2O as control or MG132 (Selleckchem, S2619) at 10 μM for 12 h to inhibit the proteasome or NH4Cl (TCI Chemicals, A2037) at 10 mM for 10 h to inhibit the lysosomal pathway; cells were collected using RIPA buffer for protein analysis. KI ESC-neurons were treated with MG132 at 10 nM, 1.0 μM, 10 μM and 100 μM at 12 h and 24 h. For protein stability assay, control cells at 0 h were collected immediately prior to CHX treatment and time profiles of 1, 3, 6, 12, 24 h were collected with freshly prepared RIPA buffer. Downstream western blot analysis was performed to calculate protein stability and half-life using Microsoft Excel and GraphPad Prism 8.

**Plasmid Constructs**. For this study, pCAGIG and pEGFP-C1 were used as base vectors. Briefly, the desired *Trim46L* or *Trim46S* isoform sequences were inserted into pCAGIG or pEGFP-C1 vectors using SacI and ApaI sites or NotI sites, respectively. Plasmids were propagated in DH5-α E. coli and prepared using Miniprep kits (Qiagen). Prior to experimentation, constructs were sequenced to ensure no mutations were introduced during the cloning process.

**CRISPR/Cas9 generation of knock-out and knock-in ES cell lines**. Mouse 46 C cells are transfected with Lipofectamine 2000 (Invitrogen) at a density of 15,000 in a 1.5 ml Eppendorf tube. For knock-out mutants, two pX330 plasmids co-expressing Cas9 enzyme and sgRNA at 0.2 μg each and 0.1 μg GFP reporter plasmid (CAG-GFP) was used to ensure exon deletion. For knock-in mutant, a single sgRNA at 0.4 μg, 1 μg of ssODN, and 0.1 μg CAG-GFP were used. Six hours post-transfection, cells are transferred to a 10-cm 0.1% gelatin-coated dish for small edited colonies to form. After 3–4 days, when colonies derived from gene editing are sufficiently large, GFP+ colonies are visualized by using fluorescent stereo-microscope to pick ~100 cell-sized colonies, treated with TrypLE Express for 3 min to dissociate into single cells, and diluted to 1 cell per well in a 96 multi-well plate to generate and screen for single-cell derived mutant colonies. Around 6–8 days later, genomic DNA is isolated and PCR genotyping along with Sanger sequencing (Eton Bioscience) is performed to confirm targeted exon deletion or knock-in of FLAG sequence in the cell line and frozen down for storage and downstream experiments.

**Neuronal Differentiation and Culture**. Mouse 46 C ESCs were maintained in 0.1% gelatin-coated 35 mm dish grown in stem cell maintenance media (DMEM/F12 (Thermo), Neurobasal (Thermo), 1x B-27 (Gibco), 1x N-2 (Gibco), 1x GlutaMAX (Thermo), 1x Pen-Strep (Fisher), and 0.05% BSA (Gibco)) with 2i inhibitors (1 μM CHIR99021-GSK inhibitor (Apexbio), 3 μM PD03259010-MEK inhibitor (Apexbio), 0.15 mM 1-thioglycerol (Sigma) and LIF (leukemia inhibitory factor (Gemini)) for 48 h prior to differentiation. ESCs were diluted to 10,000 per well in differentiation media (GMEM (Sigma), 10% Knockout Serum Replacement (KSR, Thermo), 1x GlutaMAX (Thermo), 1x non-essential amino acids (NEAA, Fisher), 1x sodium pyruvate (Gibco), and 1x Pen-Strep (Fisher)) in a U-bottom 96 well-dish (Corning, 353077) to form EBs (embryoid bodies). Additional feeding media of 100 μl was added on day 2. On day 4 after plating, EBs are transferred to 0.5% agarose-coated 10 cm dish (Greiner, 633102) in GMEM/KSR in 6 μM retinoic acid (RA). On day 6, EBs were transferred to a new 0.5% agarose-coated 10 cm dish with fresh GMEM/KSR in 6 μM RA. On day 8, GFP-Sox1 is highly expressed and is a marker for NPCs. EBs were dissociated using TrypLE Express and mechanically in a water bath at 42 °C for 10 min. Subsequently, 0.5x N-2/B-27 was added to cells

for further trituration, and cells were twice strained through a 40 μm mesh and spun down at 150 g for 5 min. Post-strain, dissociated cells were plated in 24-well for immunostaining and RNA or protein collection. DIV 1–2 cells are considered NPCs (neural progenitor cells) but onwards from DIV2 cells are considered post-mitotic neurons and feeding media is changed completely to neurobasal based B-27.

**Immunofluorescent staining**. Stem cell differentiated neurons were plated on glass coverslips (Thermo Fisher, 50949009) pre-treated with coating media (0.1 mg/mL poly L-lysine, 5 μg/mL Laminin in H2O). Cells were collected between DIV 3 and DIV 12. At the time of processing, cells were washed twice with cold PBS, fixed with 4% PFA for 15 min at room temperature, and then rinsed three times with PBS. For subsequent immunofluorescent staining, cells were washed three times with PBS, permeabilized in 0.3% Triton X-100 in PBS for 10 min, and incubated in blocking buffer (5% donkey serum, 2% BSA, 0.1% Triton X-100 in pH 7.4) for 60 min. Subsequently, cells were incubated with primary antibodies for polyclonal rabbit anti-TRIM46 (gift from Hoogenraad lab, 1:2000), monoclonal mouse anti-Tau1 clone PC1C6 (Millipore, MAB3420, 1:1000), polyclonal chicken anti-MAP2 (Abcam, ab5392, 1:2000), monoclonal mouse anti-AnkG clone N106/36 (neuroMab, 75–146, 1:250), polyclonal guinea pig anti-AnkG (Synaptic Systems, 386005, 1:250), polyclonal rabbit anti-vGLUT1 (Synaptic Systems, 135302, 1:1000), and monoclonal mouse anti-GAD67 clone 1G10.2 (Millipore, MAB5406, 1:2000) in blocking buffer at 4 °C overnight. On the next day, cells were rinsed with 0.1% Triton X-100 in PBS three times, incubated with appropriate Alexa Fluor secondary antibodies (Life Technologies, A21206, A21202, A10037, A31571, A11075, A11041, 1:2000) in PBST for 1 h at room temperature. Next, cells were stained with DAPI in PBS (1:1000) for 10 min at room temperature and later were washed three times with PBS and mounted using ProLong Gold Antifade Mount (Thermo Fisher, P36930). Images were captured on a Nikon Eclipse Ci microscope with Nikon DS-Qi2 camera and PRIOR Lumen 200 light source.

**Quantitative imaging analysis**. Images for ESC-derived neurons were acquired on the Nikon Eclipse Ci microscope at 40x magnification and were analyzed with NIS-Elements BR4.5 software. Singular neurons within the field of view (FOV) with uninterrupted and non-overlapping processes are quantified. Clustered or interconnected neurons are ignored. TRIM46 and AnkG quantification are based on their respective fluorescence intensity profiles. Neurons with no detectable TRIM46 or AnkG immunostaining at the proximal axon or AIS were categorized as TRIM46 negative (TRIM46−) or AnkG negative (AnkG−), respectively. For WT and E8-KO AnkG analysis, only TRIM46+ neurons were included in the calculation of % AnkG+ neurons. To analyze for TRIM46 specific effect on AnkG, TRIM46− (negative) cells are not included in the count. For quantification of % AnkG-positive neurons, all cells were manually counted and tallied in Excel. The same set of neurons used for measuring % AnkG+ neurons was also used to measure AnkG length and expression. To measure AnkG length, neurons must have clearly identifiable start and endpoints. The start point is the region with significant fluorescent intensity detected away from the cell body and must be continuous along the projection; the endpoint is weakened signal to the point where it could no longer be discerned from background level. AnkG length was measured by using the line tool from the NIS-Elements software and by carefully tracing along the projection of the AnkG+ neuron. For AnkG intensity, an outline for region of interest (ROI) is used to trace the proximal axon or AIS region for measurement. The ROI fluorescent AnkG intensity is subtracted from background to get the corrected value and this corrected value is used as normalized mean AnkG intensity. TRIM46 length and normalized mean intensity for WT and E8-KO neurons were measured and quantified the same as AnkG. For E10-KO analysis, AnkG length and intensity were measured under the same procedure. For vGLUT1 and GAD67 analysis, only DAPI+ neurons were included in calculation. vGLUT1+ neurons exhibited easily identifiable and distinct signal whereas GAD67+ neurons must be non-vGLUT1+ and show specific and elevated staining above background levels.

**Axonal polarity and axon number**. For Tau1-based polarity analysis, the mean dendritic intensity ($I_d$) in dendritic regions and mean axonal intensity ($I_a$) in axonal regions was used to calculate the axonal polarity index $P = \frac{I_a - I_d}{I_a + I_d}$. $P > 0$ means polarized axonal distribution; $P < 0$ means polarized dendritic distribution; $P = 0$ ($I_a = I_d$) means uniform distribution. The number of axons were defined based on Tau1 staining. ESC-neurons lacking specific Tau1 staining in the distal axon were categorized as 0 axon whereas single axon (1) and double axons (2) were identified by distinct Tau1 signal in projections.

**Protein collection and western blot analysis**. N2a cells or ESC-derived neurons were lysed in freshly prepared RIPA buffer containing 10 mM Tris-HCl pH 7.4., 150 mM NaCl, 1 mM EDTA, 1% Triton X-100, 0.1% SDS, 1 mM PMSF, 100 U/ml Turbonuclease and 1x protease inhibitor cocktail (Roche) as previously described[43]. Protein lysates were cleared by centrifugation at 18,800 g for 10 min at 4 °C using a tabletop centrifuge to separate insoluble fractions. To determine protein concentration, the BCA assay (Thermo Scientific, 23227) was performed using BSA dilutions as the protein standard. Samples were then boiled with 4x

Laemmli buffer (Biorad, 1610747) containing 10% β-mercaptoethanol and then denatured at 95 °C for 10 min. Equal protein amounts were loaded onto were 10% SDS-PAGE gels for western blot. Proteins from SDS-PAGE gel were transferred to Immobilon-FL PVDF membranes (Millipore, IPFL00010) by semi-dry transfer (Biorad, 1703940). Transferred proteins on PVDF were blocked using 1% BSA in TBSTween-20 (0.1%) for one hour prior to overnight antibody incubation. Primary antibodies include polyclonal rabbit anti-TRIM46 (Hoogenraad lab, 1:1000), polyclonal rabbit anti-TRIM46 (Proteintech, 21026-1-AP, 1:1000), monoclonal mouse anti-α-TUBULIN clone DM1A (Calbiochem CP06–100μg, 1:100), polyclonal mouse anti-GAPDH (Thermo Scientific, AM4300, 1:2000), polyclonal rabbit anti-FLAG (Sigma-Aldrich, F7425, 1:2000), polyclonal rabbit anti-UBIQUITIN (Cell Signaling, 3933, 1:1000), and polyclonal chicken anti-GFP (Aves, GFP-1020, 1:3000). Primary antibodies were detected by appropriate Alexa Fluor conjugated secondary antibodies (Thermo Fisher; A21206, A10042, A31573, A11039, A10037, A31571, 1:2000) or HRP linked antibody (Cell Signaling, 7074, 1:5000). Blots were visualized using Typhoon FLA 9000 (fluorescence) or ChemiDoc (chemiluminescence via Radiant Plus substrate, AC2103). Band intensities were quantified using ImageQuantTL and α-TUBULIN was used to derive normalized TRIM46 protein levels.

**Immunoprecipitation**. To detect endogenous TRIM46 or FLAG-TRIM46 protein isoforms by immunoprecipitation, E12.5 brain, P10 cortices, or ESC-derived neuron lysate are collected with IP buffer (50 mM Tris, pH 7.4, 150 mM NaCl, 1 mM EDTA, 0.10% Triton X-100, 1 mM PMSF, 1x protease inhibitor cocktail (Roche), 100 U/ml Turbonuclease (Sigma)). Lysate is cleared through centrifugation at 18,800 g for 15 min in 4 °C. Protein A magnetic beads (Biorad, 1614013) or FLAG magnetic beads (Sigma-Aldrich, M8823-5ML) are washed three times with IP buffer. After washing, beads are incubated at 20 rpm at RT with either rabbit TRIM46 antibody (Proteintech, 21026-1-AP) or rabbit anti-IgG control antibody (Proteintech, 30000-0-AP) for 20 min to form bead-antibody complexes. For FLAG beads, no incubation is needed. After the bead-antibody incubation step, complexes are washed three times (same as lysis buffer) to remove unbound antibodies. After centrifugation, cleared lysate is split equally half and half to control IgG-IP and TRIM46-IP. Incubation is performed at 25 rpm in 4 °C for 4 h with the first flowthrough (FT) for FLAG-IP is collected and beads are washed three times. Bead bound proteins were denatured by boiling in 2x Laemmli buffer with β−ME at 95 °C for 10 min. Using a magnetic plate, immunoprecipitated proteins are separated from beads and ready for western blot analysis. The IP fractions were resolved by SDS-PAGE, transferred to Immobilon-FL PVDF membrane, and probed with primary antibodies that include polyclonal rabbit anti-TRIM46 (Proteintech, 21026-1-AP, 1:1000) or polyclonal rabbit anti-FLAG (Sigma-Aldrich, F7425, 1:2000). Primary antibodies were detected by appropriate Alexa Fluor conjugated secondary antibodies (Thermo Fisher; A21206, A10042, A31573, 1:2000) and blots were visualized using Typhoon FLA 9000.

**Mouse care**. Mice were maintained in accordance with the guidelines of the Institutional Animal Care and Use Committees at the University of California, Riverside (UCR). Under routine monitoring and checking by veterinary and mouse room staff, all animals used in this study were kept healthy in a mouse room with a stable temperature at 22 ± 2 °C and in a 12 h light/dark cycle. Mice were maintained in ventilated cages and clean bedding was provided every two weeks. Food and water were provided ad libitum. No more than 5 mice were allowed in each cage. Mice of both sexes were used. The $Ptbp2^{loxp/loxp}$ mice[32] and $Upf2^{loxp/loxp}$ mice[40,55] used in this study were previously described. Briefly, $Ptbp2$ conditional KO mice ($Ptbp2$-cKO) were generated by breeding $Ptbp2^{loxp/loxp}$ with $Ptbp2;^{loxp/+}$Emx-Cre. Additionally, $Upf2$ conditional KO mice ($Upf2$-cKO) were generated by breeding $Upf2^{loxp/loxp}$ with $Upf2;^{loxp/+}$Emx-Cre mice. All animal procedures were approved by the UCR Institutional Animal Care and Use Committee.

**Mouse primary neuron culture**. Prior descriptions of culturing primary cortical neurons have been reported[32,40]. Mouse neocortices were dissected at targeted E13.5 stage and dissociated in 0.25% trypsin (Thermo Scientific) comprising of 1 mM EDTA (Sigma) in 1x HBSS (Invitrogen) at 37 °C for approximately 8 to 10 min. Subsequently, dissociated neurons are plated in fresh media containing 25 mM glucose (Sigma), 1x GlutaMAX (Thermo), and 20% heat-inactivated horse serum (Thermo) in MEM (Invitrogen). After 6 h, plating media is replaced with neurobasal-based feeding media consisting of 1x B-27 (Invitrogen) and 1x Gluta-MAX. For CHX treatment (Fisher Scientific), neurons one day later (E13.5-DIV 1) were treated at final concentration of 0.5 mg/ml for 5 h and collected in TRizol for RNA extraction.

**RNA-Seq data analysis**. Mouse adult tissue RNA-seq data (generated by Thomas Gingera's lab) was downloaded from ENCODE. We mapped the RNA-seq data using STAR based on the mouse GENCODE annotation (mmv20). We obtained exon junction reads for *Trim46* exon 8 and exon 10 and calculated inclusion ratio (PSI) as the following: 0.5 x inclusive junction reads / (0.5 x inclusive junction reads + exclusive junction reads) x 100. Exclusion ratio (%) is defined as 100−PSI. Exon 8− isoform expression level is determined as steady-state expression TPM x exon 8 exclusion ratio. Exon 10+ isoform expression level is determined as steady-

state expression TPM x exon 10 inclusion ratio. The Gingera dataset contains two biological replicates for each tissue. PSI and TPM values are presented as the average of two biological replicates. Tissue specificity index $\tau =$

$\frac{\sum_{i=1}^{n}(1-\hat{x}_i)}{n-1}$, where $\hat{x}_i = \frac{x_i}{\max\limits_{1\le i\le n}(x_i)}$, $n$ is the number of tissues and $x_i$ is the expression of the gene in tissue $i$, $x_i = \log_2(TPM_i + 1)$. We took the average of $\log_2(TPM_i + 1)$ values from three brain tissues to obtain one value for the brain tissue.

**Statistics**. All statistical analyses were performed using Microsoft Excel with Student's t-test under paired or unpaired parameters and under two-tailed or one-tailed conditions as indicated with the exception of half-life calculation performed using GraphPad Prism 8 software.

**Reporting summary**. Further information on research design is available in the Nature Research Reporting Summary linked to this article.

## Data availability
The data supporting the findings of this study are available from the corresponding authors upon reasonable request. Source data for the figures and supplementary figures are provided as a Source Data file. Source data are provided with this paper.

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

## Acknowledgements

We thank Dr. Casper Hoogenraad (Utrecht University, Netherlands) for the TRIM46 antibody. We thank Dr. Qilong Ying (USC) for the 46 C Sox1-GFP ES cell line. We thank Cristina Gonzalez for help with generating knockout ES cell lines. We thank Dr. Lin Lin and Dr. Min Zhang for assisting with mouse samples. This study is supported by National Institute of Health (NIH) grants R01NS104041 and R01MH116220 (S.Z.) and R01GM137428 (L.C.). Figures 2e, f, 4a, 5a, and 6a were created with assistance from Biorender.com.

## Author contributions

J.K.V. and S.Z. conceptualized the project; J.K.V. and V.E. optimized ESC differentiation protocol; V.E. supervised exons 10 and 8 KO ES cell line generation and J.K.V. assisted and verified KO through Sanger sequencing; V.E. generated the 3xFKI:*Trim46* ES cell line; J.K.V. carried out most of the experiments; L.C. performed RNA-seq data analysis; J.K.V. and S.Z. analyzed the data and wrote the paper.

## Competing interests

The authors declare no competing interests.
