## [Peer Review File · Nature Communications]

Title: Alternative Splicing, NMD, and Protein Stability Controls Synergistically Determine Temporal Induction and Tissue-specific Expression of TRIM46 during Axon FormationREVIEWER COMMENTS

Reviewer #1 (Remarks to the Author):

Vuong et al. examine the regulation of TRIM46, a cytoskeletal protein associated with regulation of the axon initial segment and axon formation. They use an in vitro ES cell differentiation model to monitor TRIM46 protein and mRNA isoform expression. They report that TRIM46 is up-regulated upon early stages of differentiation of ES cells into neurons. Interestingly, the overall TRIM46 mRNA levels are poorly predictive of protein expression. The authors propose two major mechanisms at the level of mRNA isoforms for this regulation: First, the incorporation of Exon 10 where protein (TRIM46L) derived from Exon 10 containing mRNAs exhibits much longer half-life than protein derived from Exon10-lacking mRNAs. Second, incorporation of Exon 8 which targets mRNAs to nonsense-mediated decay – thereby modifying protein abundance.

Overall, the experiments appear carefully performed and well documented. The paper is clearly written and reads well. There are some specific points that should be addressed:

Most importantly, title and conclusions refer to axon formation – but the primary read-out used is AnkG accumulation and the length of AnkG stained structures. The authors should probe the LOF phenotypes reading out actual markers of axons and axo-dendritic polarity.

Second, the severity of the loss-of-function phenotype in the ES cell-derived neuron system appears significantly less pronounced than what was previously reported in the literature. Do Exon10 ES cells differ in what type of neuron these cells differentiate? This should be quantified using markers, scRNA-Seq, or an alternative method.

Third, the interplay of exon 8 and exon 10-dependent regulatory mechanisms remains unknown. Are these truly independent? Or are exon 8 -containing mRNAs more likely to contain or lack exon 10? Which of these mechanisms is most relevant in vivo? The authors propose a “multi-layered control” – but it remains somewhat mysterious how the two mechanisms are integrated, whether they occur independently, and what is their respective contribution.

Specific/Additional points:

- 1) An important starting point for the investigations is the up-regulation of TRIM46 from EB – DIV1 to DIV3 – DIV 7 stages of neuronal differentiation. The text refers to observations that TRIM46 “was not detected during EB stages or DIV1” – please include this data in the manuscript (Fig.1c).
- 2) To allow for assessing the correlation (or lack of correlation) of protein and mRNA levels the authors should run qPCR probing total Trim46 mRNA levels over EB 8 – DIV7.
- 3) In the experiments on FLAG-TRIM46S protein stability it would help to probe whether mRNA levels altered or not. This would help concluding whether protein stability is the primary mechanism for lack of protein – or whether additional factors (translational regulation?) might be involved.
- 4) Considering that the authors generated 3xFLAG KI cells they should use these to probe whether the

endogenous (rather than overexpressed) TRIM46 protein is regulated by proteasomal degradation.

5) The authors should provide quantitative information on the total TRIM46 mRNA levels in Exon 8 and Exon 10 knock-out cells (qPCR).

6) What happens to Exon10 insertions in Exon 8 KO and vice versa. mRNA isoform analysis for Exon 8 and Exon10 deletion cells should be provided to confirm that no cryptic splice sites modify the resulting transcripts.

Minor point:

The authors state: “no tissues other than the brain form axons” – there are axons in spinal cord, peripheral sensory neurons, and enteric nervous system. Better refer to “neuronal tissue” or similar.

Reviewer #2 (Remarks to the Author):

In this manuscript, Vuong et al. tested an intriguing possibility that the emergence of TRIM46, one of the earliest proteins accumulated in the axon initial segment (AIS), is regulated by alternative splicing. The authors combined a well described embryonic stem cell-based neural lineage differentiation system with CRISPR/Cas9 knockout and knockin approach to examine the biochemical and functional consequences of alternative splicing of TRIM46. They showed that inclusion or exclusion of exon 10 or exon 8 could affect the production of a stable form of TRIM46 protein, which in turn impact AnkG localization in the AIS. Overall, this study was carefully done, and the data revealed a novel mechanism involving specific alternative splicing events in the control of expression of axon initiation proteins. I only have a few minor comments.

1. For Figure 1a, spell out GMEM/KSR and RA in figure legend.

2. Add DNA size labels to Figure 1 (panels f and g) and Figure 5 (panels b, c, e, and g).

3. Figure 7 showed a nice tissue-specific expression profile of TRIM46 in relation to splicing of exon 10 or exon 8. The authors can use embryonic brain data generated by the ENCODE project to assess the temporal expression profile of TRIM46 and splicing choices of exon 10 and exon 8 during brain development. Presumably, increasingly higher TRIM46 mRNA levels would be positively correlated with inclusion of exon 10 or exclusion of exon 8 from early to late stages of neurogenesis.

Reviewer #3 (Remarks to the Author):

Nature communications,
NCOMMS-21-12859

“Multi-layered Integrated Regulations Control Temporal Induction and Tissue-specific Expression of TRIM46 for Axon Formation”

In this manuscript the authors describe how expression of TRIM46, an important determinant in axonogenesis, is regulated at two mechanistic levels, involving NMD-mediated mRNA degradation (skipping/inclusion of exon 8) and protein stability (exon 10 skipping/inclusion). Both cell culture and mouse KO/KI-strategies are used, in combination with a neuronal differentiation system. The study is carefully done, well documented and presented with solid data, and the results are clearly described in text/figures and overall convincing. General interest should be high for both RNA-processing and neurobiology communities. I have several comments and suggestions/requests for modifications:

1. The title should be toned down somewhat (“multi-layered integrated” ?).
2. When focussing on exon 8 and 10 alternative splicing, it should be made clear why these were selected within the TRIM46 gene.
3. The same applies to the focus on PTBP2 as the only RNA-binding protein they analyze. Obviously it is established as an important brain-specific regulator, but were other candidate RBPs considered? Old CLIP data from the Darnell lab were used (Licatalosi, 2012; here: Figure 1i). At least in the supplement the entire gene should be shown in its iCLIP tag pattern, not only the exon 10 region. Since these old data were from whole mouse brain, CLIP should be repeated in their differentiation system to allow better comparison and correlation with the underlying pattern of alternative splicing changes.
4. Results, p.11 top line: Since at least in some experiments low residual levels of the short protein isoform can be detected (Figure 2j), “below detection sensitivity” should be changed.
5. Figure 3cd: Were these quantitations of protein levels after CHX treatment normalized?
6. Figures 5/6, exon 8 as an NMD target. This is well demonstrated by CHX treatment, as well as UPF2-KO and UPF1 siRNA-knockdown, and by conservation data. It would be even more convincing if another exon of TRIM46 were included as a negative control.
7. Did the authors make an attempt to identify what exon 10-dependent differences on the protein level and protein regions determine protein destabilization?

We are pleased that the reviewers recognize the novelty and significance of our work, and would like to thank the reviewers for their constructive comments to strengthen the manuscript and. The insightful suggestions raised by the reviewers are indicative of how carefully they considered the presented data, and we are grateful for the time and effort this took. In pursuing the recommended experiments, we have been able to deepen our understanding, validate and strengthen our findings. We have added substantial new data and figures (Figures 4e, 6h, Supplementary Figures 1b, 2, 3, 6a, 7, 8, 9c-f, 10, 11, 12) in the revision, which significantly strengthen the manuscript.

Responses to individual reviewer's comments are presented below and integrated into the main text of the revised manuscript (line numbers provided). As requested, these changes are highlighted in the manuscript file. We hope the reviewers find our revised manuscript appropriate for publication at Nature Communication.

Reviewer #1 (Remarks to the Author):

Vuong et al. examine the regulation of TRIM46, a cytoskeletal protein associated with regulation of the axon initial segment and axon formation. They use an in vitro ES cell differentiation model to monitor TRIM46 protein and mRNA isoform expression. They report that TRIM46 is up-regulated upon early stages of differentiation of ES cells into neurons. Interestingly, the overall TRIM46 mRNA levels are poorly predictive of protein expression. The authors propose two major mechanisms at the level of mRNA isoforms for this regulation. First, the incorporation of Exon 10 where protein (TRIM46L) derived from Exon 10 containing mRNAs exhibits much longer half-life than protein derived from Exon10-lacking mRNAs. Second, incorporation of Exon 8 which targets mRNAs to nonsense-mediated decay – thereby modifying protein abundance.

Overall, the experiments appear carefully performed and well documented. The paper is clearly written and reads well. There are some specific points that should be addressed:

Most importantly, title and conclusions refer to axon formation – but the primary read-out used is AnkG accumulation and the length of AnkG stained structures. The authors should probe the LOF phenotypes reading out actual markers of axons and axo-dendritic polarity.

Response: we thank reviewer #1 for the positive comments. We understand reviewer #1's point of view. Axon formation can be a broadly used term encompassing many molecular cellular events leading to the ultimate generation of axon. These can include, but are not limited to, axonal growth, axonal specification, establishment of axo-dendritic polarity and the compartmentalization of axon-specific molecules (axonal markers). AnkG is enriched in the axon but not dendrites, hence is widely used as an axonal marker in certain contexts. We chose to use AnkG as the major readout because both AnkG and TRIM46 are enriched at the AIS in axons.

To address the reviewer's comment about additional axonal marker and polarity, we stained the knockout neurons with axonal marker Tau1. We then quantified its average dendritic intensity (I_d) in dendritic regions and average axonal intensity (I_a) in axonal regions. We calculated the

axonal polarity index $P = \frac{Ia - Id}{Ia + Id}$. $P > 0$ means polarized axonal distribution; $P = 0$ means uniform distribution; $P < 0$ means polarized dendritic distribution. As shown in Supplementary Figures 8b, control neurons show $P_{\text{Tau1}} = 0.767$ while exon 10 knockout neurons exhibit $P_{\text{Tau1}} = 0.765$. Furthermore, most neurons possess a single axon in both control (86.4%) and exon 10 KO neurons (85.9%). There is essentially no difference among WT and mutants regarding the percentage of neurons with 2 axons (7.8% and 8.6% for control and exon 10 KO, respectively, $p=0.83$, two-tailed, unpaired t-test) or without axons (5.8%, 5.5% for control and exon 10 KO respectively, $p=0.87$, two-tailed unpaired t-test). Since Tau1 polarity and axon number are unaffected, the exon 10 knockouts do not appear to alter the initial step of axonal polarization but the subsequent formation of AnkG cluster. Therefore, temporal control of *Trim46* splicing and expression regulates certain but not all aspects of axon formation. (lines 359-373)

To future address reviewer's point, we have changed the title: "Multilayered Regulations of Alternative Splicing, NMD, and Protein Stability Control Temporal Induction and Tissue-specific Expression of TRIM46 during Axon Formation".

Second, the severity of the loss-of-function phenotype in the ES cell-derived neuron system appears significantly less pronounced than what was previously reported in the literature. Do Exon10 ES cells differ in what type of neuron these cells differentiate? This should be quantified using markers, scRNA-Seq, or an alternative method.

Response: Reviewer 1 raised an interesting question and a possible explanation. van Beuningen et al. reported the *Trim46* knockdown (by shRNA transfection) in cortical and hippocampal cultures without indicating the neuronal subtypes of the knockdown (or transfected) neurons. Their cultures presumably contained largely glutamatergic neurons but also GABAergic neurons. Because previous works did not provide information about the cell type, it is impossible for us to compare apple to apple.

The ES cell differentiation protocol we adopted was validated by multiple groups to produce glutamatergic neurons (PMID: 18983967, 26293963, 21094123, 23988668, 16696855, 24940564). We examined the neuronal subtypes derived from WT and exon 10 knockout ES cell-derived neurons. We found no differences in population composition between WT and E10-KO ESC-neurons: the overwhelming majority are 97% glutamatergic neurons for both (Supplementary Fig. 8d and 8e). Given similar percentage of vGLUT1+ and GAD67+ cells between control and exon 10-KO neurons this shows the LOF phenotype is not due to alteration in neuronal populations. (lines 374-380)

There are other possible explanations for the "less pronounced" phenotype. First, the timing and kinetics of TRIM46 deletion. van Beuningen et al.'s isolated hippocampal cultures and cortical cultures from E18 rat brains, and transfected the cultures with shRNA at DIV 1 and cortical cultures at DIV 4. Because shRNA typically takes 2 or more days to deplete proteins (depending on shRNA efficiency and protein stability), it is likely TRIM46 is already well expressed and assumes functions of controlling AIS formation before being knocked down. In other words, van Beuningen's method acutely cuts off the supply of TRIM46 in the midst of exerting its cellular function. In E10-KO neurons, TRIM46 is not detectable at any time. Although

this may sound worse than the knockdown, the absence may trigger compensatory mechanisms earlier, which provides neurons more time to adjust to the loss of TRIM46, e.g., by inducing the expression and accumulation of TRIM46-like molecules earlier. Given that TRIM46 is not absolutely required for AnkG accumulation, functionally redundant proteins likely exist to compensate in the absence of TRIM46. (lines 601-616)

Second, RNAi-mediated knockdown is well known to have secondary effects including off-targets, which could worsen the phenotype. Third, although transfection alone does not affect axon formation. The stress caused by transfection can sensitize neurons to the depletion of TRIM46, exacerbating the phenotype. We have added these possibilities in Discussion.

Third, the interplay of exon 8 and exon 10-dependent regulatory mechanisms remains unknown. Are these truly independent? Or are exon 8 -containing mRNAs more likely to contain or lack exon 10? Which of these mechanisms is most relevant in vivo? The authors propose a “multi-layered control” – but it remains somewhat mysterious how the two mechanisms are integrated, whether they occur independently, and what is their respective contribution.

Response: Based on the comment, Reviewer 1 probably refers to regulation of exons 8 and 10 splicing as two regulatory mechanisms when asking “how the two mechanisms are integrated, whether they occur independently...”. To clarify, multilayered controls in our manuscript refer to the following three layers: alternative splicing control, NMD control of RNA stability, and protein stability control. We propose coupling of these vertical regulatory layers. Specifically, alternative splicing of exon 8 is coupled to NMD control of *Trim46* transcripts. Alternative splicing of exon 10 is coupled to protein stability control of TRIM46 proteins. Regulations of exon 8 and exon 10 both belong to the layer of alternative splicing control and we did not propose their horizontal integration.

However, reviewer 1 did raise an important question regarding the relationship between exon 8 and exon 10 splicing. This is a legit question since temporal splicing of exons 8 and 10 is concurrent during neuronal differentiation. So, the question is whether this concurrent splicing (association) indicates dependency (causation) or coordinated regulation.

If splicing of exons 8 and 10 are independent, neurons can express four *Trim46* isoforms (E8+E10-, E8+E10+, E8-E10-, E8-E10+). If splicing of exons 8 and 10 are independent, neurons can express four *Trim46* isoforms (E8+E10-, E8+E10+, E8-E10-, E8-E10+). If one exon splicing fully determines the other, neurons would express only two of these four isoforms (E8+E10- and E8-E10+), since exon 8 inclusion is correlated with exon 10 skipping (Fig. 1f and 5b). To distinguish these two possibilities, we designed primers to amplify regions from exon 7 to exon 11. As shown in Supplementary Fig. 11a, we observed all four isoforms in all developmental stages. (lines 473-479)

To answer the question “are exon 8-containing mRNAs more likely to contain or lack exon 10”, we calculated the percentage of exon 10 inclusion in exon 8-containing mRNAs (E8+E10+/E8+) and exon 8-skipping mRNAs (E8-E10+/E8-). The percentage of exon 10 inclusion in exon 8-containing mRNAs (E8+E10+/E8+) increased dramatically during axon formation rather than staying low (8.1%, 36%, 56%, 65% at EB D8, DIV 1, 3, 5, 7, respectively, Supplementary Fig.

11b). That is, exon 8-containing mRNA did not necessarily tend to skip exon 10. Instead, it contained or lack exon 10 depending on the differentiation status. Exon 8-containing mRNA tended to lack exon 10 at EB D8 but gradually increased exon 10 inclusion till DIV 7, just like exon 8-skipping mRNA (Supplementary Fig. 11b). (lines 479-485)

To directly test causality, we asked whether exon 8 knockout affects exon 10 splicing, and vice versa. We found exon 8 inclusion did not change in exon 10-KO neurons in comparison to control ES-derived neurons during differentiation (Supplementary Fig. 11c and 11d). Similarly, exon 10 inclusion was not significantly affected in E8-KO neurons throughout neuronal differentiation from EB stage to maturing neurons at DIV 7 (Supplementary Fig. 11e and 11f). We also note that in PTBP2-cKO brains, exon 10 splicing changes (Fig. 1g-1i), but exon 8 splicing does not (Supplementary Fig. 12a and 12b), showing that exon 10 splicing does not affect exon 8 splicing. Therefore, exon 10 splicing itself is not a determinant of exon 8 splicing, and vice versa. (lines 487-495)

Another possibility for concurrent splicing is common upstream regulators, i.e., exons 8 and 10 are coordinated by the same RNA binding protein(s). Reviewer 1 asks “Are these truly independent?”. If “truly independent” implies completely different upstream regulators, one can never prove independent regulation of the two exons, because there is always possibility of unknown common regulators for the two exons. We found PTBP2 inhibits exon 10 (Fig. 1g and 1i) but not exon 8 (Supplementary Fig. 12a, 12b). Although this does not exclude the possibility of other regulators that control both exons, it does show the two exons can be regulated by different factors, at the very least one by PTBP2 and the other not.

Taken all together, our data suggest that exons 8 and 10 splicing are likely coordinated by the differentiation signal upstream of their trans-regulators. The signal (either intrinsic or extrinsic or both) likely influence two different (but possibly overlapping) sets of *trans* regulators that each control exon 8 and exon 10 splicing. (lines 494-497)

To answer “Which of these mechanisms is most relevant in vivo” would require generation of exon 8 and exon 10 knockout animals. And even then, it is not clear how to benchmark the comparison, because exon 8 knockout upregulates TRIM46 proteins whereas exon 10-knockout depletes TRIM46 proteins. We therefore respectfully think this is beyond the scope of this paper. Our CRISPR-knockout experiments shows both splicing events are important for controlling TRIM46 expression. We do not make conclusions on which exon is more important but show that the regulation of two alternative splicing events converges to induce functional TRIM46 proteins.

Specific/Additional points:

1) An important starting point for the investigations is the up-regulation of TRIM46 from EB – DIV1 to DIV3 – DIV 7 stages of neuronal differentiation. The text refers to observations that TRIM46 “was not detected during EB stages or DIV1” – please include this data in the manuscript (Fig.1c).

Response: We previously ran Western blots of TRIM46 proteins at EB D8 and DIV 1 along with some but not all late differentiation stages. To address the reviewer's comment, we now ran samples from EB D8 to DIV 7 and added this data to Supplementary Fig. 1b. The observations have been consistent. We kept the original Fig. 1c for the purpose of being consistent with the quantification plot (Fig. 1d).

2) To allow for assessing the correlation (or lack of correlation) of protein and mRNA levels the authors should run qPCR probing total *Trim46* mRNA levels over EB 8 – DIV7.

Response: we have performed RT-qPCR probing total *Trim46* mRNA levels using qPCR primers amplifying regions of constitutive exons 5 and 6. As shown in Fig. 4e and 6h, we found via RT-qPCR total *Trim46* mRNA levels in WT ES-cell derived neurons are upregulated from EB day 8 to DIV 7. Similarly, exons 8 and 10 KO-neurons also exhibit transcriptional increase in total mRNA over the same development stages. Interestingly, steady-state mRNA levels of E8-KO neurons on EB day 8 shows substantial upregulation over WT neurons (Fig. 6h). This is because in WT neurons exon 8 is largely included at EB day 8 leading to NMD of *Trim46* transcripts. Exon 8 deletion prevents NMD of *Trim46* transcripts, effectively increasing its steady-state abundance. Therefore, this data confirms the significant role of exon 8 splicing and NMD in regulating *Trim46* transcript abundance. (lines 438-446)

This data further supports multi-layered controls of *Trim46* expression incorporating transcriptional regulation. The loss of TRIM46 proteins was not due to transcriptional defects in E10-KO because there was no significant difference in total *Trim46* mRNA levels between control and mutant neurons (Fig. 4e). Or, transcriptional regulation alone is not responsible for the change in TRIM46 protein expression in the mutant neurons. Indeed, the lack of functional TRIM46 proteins in exon 10 KO demonstrates that exon 10 splicing regulation is an essential gatekeeper of TRIM46 protein function. (lines 332-340)

3) In the experiments on FLAG-TRIM46S protein stability it would help to probe whether mRNA levels altered or not. This would help concluding whether protein stability is the primary mechanism for lack of protein – or whether additional factors (translational regulation?) might be involved.

Response: Reviewer #1 asked a good question. We used RT-qPCR to measure mRNA levels of FLAG-*Trim46L* and FLAG-*Trim46S* transcripts under the same experimental condition as for Fig. 3a and b. We observed an increase in mRNA levels of two FLAG-*Trim46* isoforms with increasing plasmid dosage from 0.1 to 1.0 μ g but importantly, the mRNA levels showed no difference between the two FLAG-*Trim46L* and FLAG-*Trim46S* transcripts. This is shown in Supplementary Fig. 6a. (lines 264-267). Please note that the half-life of TRIM46S protein is 1.9 hours, whereas that of TRIM46L is 4.3 hours (Fig. 3c and d). Stabilization of TRIM46S proteins is also more sensitive to proteasome inhibitor MG132 than that of TRIM46L proteins (Fig. 3e and f). These pieces of evidence, taken together, show that protein stability is a significant regulatory mechanism.

4) Considering that the authors generated 3xFLAG KI cells they should use these to probe whether the endogenous (rather than overexpressed) TRIM46 protein is regulated by proteasomal degradation.

Response: Reviewer has a good point. We note that TRIM46L protein expressed from 0.2 µg plasmid transfection is comparable to endogenous TRIM46 levels in DIV 5-7 neurons. In Fig. 3, we did various transfection doses and determined the lowest dose for the following MG132 and NH4Cl experiments and show that the lack of TRIM46S protein detection in comparison to TRIM46L protein can be mimicked by expressing low amounts of exogenous TRIM46 proteins.

We did the experiment as the reviewer suggested. In multiple attempts, the ES cell-derived 3xFLAG-KI neurons did not exhibit clear proteasome inhibition in response to MG132 treatment before showing cytotoxicity. MG132 was typically used in a range from 10 nM to 100 µM. For example, N2a cells treated with 10 µM MG132 for 12 hours showed a markedly increase in ubiquitinated proteins (Supplementary Fig. 7a). However, in 3XFLAG-KI neurons treated with MG132 from 10 nM to 100 µM there was little change in ubiquitinated proteins (Supplementary Fig. 7a). We also extended the treatment time to 24 hours, which did not upregulate the ubiquitin signals but did cause massive cell death at various concentrations (Supplementary Fig. 7b). Unfortunately, the MG132 toxicity (probably proteasome-independent) in ESC-derived neurons precluded further investigation. (lines 300-305)

5) The authors should provide quantitative information on the total TRIM46 mRNA levels in Exon 8 and Exon 10 knock-out cells (qPCR).

Response: We used RT-qPCR to measure total *Trim46* mRNA levels in Exon 8 and Exon 10-KO cells. Please see the response to point 2.

6) What happens to Exon10 insertions in Exon 8 KO and vice versa. mRNA isoform analysis for Exon 8 and Exon10 deletion cells should be provided to confirm that no cryptic splice sites modify the resulting transcripts.

Response: This is an important quality control experiment. As suggested, we performed RT-PCR for exon 8 splicing in exon 10 KO-cells and exon 10 splicing in exon 8-KO cells at EB day 8, DIV 1, and DIV 7. We did not detect any cryptic isoforms (Supplementary Fig. 11c and 11e). There were two weaker bands between 300 bp and 350 bp in the analyses of exon 8 splicing, which constantly showed up as in the WT ESC-derived neurons (Fig. 5b). We note that these two bands typically correlate with the abundance of exon 8 plus transcripts and intensify with increasing PCR cycles, suggesting they are likely heteroduplex of exon 8+ and exon 8- amplicons. Given the small size of exon 8 (38 bp), this is very likely. More importantly, we found exon 8 inclusion did not change in exon 10-KO neurons in comparison to control ES-derived neurons during differentiation (Supplementary Fig. 11c and 11d). Similarly, exon 10 inclusion is not affected in E8-KO neurons throughout neuronal differentiation from EB stage to maturing neurons at DIV 7 (Supplementary Fig. 11e and 11f). (lines 486-497)

Minor point:

The authors state: “no tissues other than the brain form axons” – there are axons in spinal cord, peripheral sensory neurons, and enteric nervous system. Better refer to “neuronal tissue” or similar.

Response: We thank Reviewer #1’s comment. We have corrected this (line 499).

Reviewer #2 (Remarks to the Author):

In this manuscript, Vuong et al. tested an intriguing possibility that the emergence of TRIM46, one of the earliest proteins accumulated in the axon initial segment (AIS), is regulated by alternative splicing. The authors combined a well described embryonic stem cell-based neural lineage differentiation system with CRISPR/Cas9 knockout and knockin approach to examine the biochemical and functional consequences of alternative splicing of TRIM46. They showed that inclusion or exclusion of exon 10 or exon 8 could affect the production of a stable form of TRIM46 protein, which in turn impact AnkG localization in the AIS. Overall, this study was carefully done, and the data revealed a novel mechanism involving specific alternative splicing events in the control of expression of axon initiation proteins. I only have a few minor comments.

Response: We thank Reviewer #2 for the positive comments and recognizing the novelty of our work.

1. For Figure 1a, spell out GMEM/KSR and RA in figure legend.

Response: As suggested, we have spelled them out in the figure legend (lines 1048-1051).

2. Add DNA size labels to Figure 1 (panels f and g) and Figure 5 (panels b, c, e, and g).

Response: Based on recommendation, we have added DNA size labels to the figures.

3. Figure 7 showed a nice tissue-specific expression profile of TRIM46 in relation to splicing of exon 10 or exon 8. The authors can use embryonic brain data generated by the ENCODE project to assess the temporal expression profile of TRIM46 and splicing choices of exon 10 and exon 8 during brain development. Presumably, increasingly higher TRIM46 mRNA levels would be positively correlated with inclusion of exon 10 or exclusion of exon 8 from early to late stages of neurogenesis.

Response: We thank Reviewer 2’s comment. As suggested by the reviewer, we analyzed the embryonic brain data generated by ENCODE/Cold Spring Harbor lab (GSE36025) specifically for the temporal mRNA expression of *Trim46* mRNA and splicing of exons 8 and 10. The ENCODE data from Caltech have significant less junction reads and are not used. As shown in Supplementary Fig. 2, exon 10 inclusion gradually increases, and exon 8 inclusion decreases from E11 to E18, which is consistent with increasingly higher levels of total *Trim46* mRNA levels during the same period.

Reviewer #3 (Remarks to the Author):

Nature communications,
NCOMMS-21-12859

“Multi-layered Integrated Regulations Control Temporal Induction and Tissue-specific Expression of TRIM46 for Axon Formation”

In this manuscript the authors describe how expression of TRIM46, an important determinant in axonogenesis, is regulated at two mechanistic levels, involving NMD-mediated mRNA degradation (skipping/inclusion of exon 8) and protein stability (exon 10 skipping/inclusion). Both cell culture and mouse KO/KI-strategies are used, in combination with a neuronal differentiation system. The study is carefully done, well documented and presented with solid data, and the results are clearly described in text/figures and overall convincing. General interest should be high for both RNA-processing and neurobiology communities. I have several comments and suggestions/requests for modifications:

Response: We thank Reviewer 3 for the positive comments and recognizing the impact of our discoveries.

1. The title should be toned down somewhat (“multi-layered integrated” ?).

Response: We have toned down the title by removing “integrated”. Multi-layered controls refer to the layers of alternative splicing control, NMD control of mRNA levels, and protein stability control. We propose integration of these vertical regulatory layers that converge on controlling the timing of TRIM46 induction. Specifically, alternative splicing of exon 8 is coupled to NMD control of *Trim46* transcripts. Alternative splicing of exon 10 is coupled to protein stability control of TRIM46 proteins.

2. When focusing on exon 8 and 10 alternative splicing, it should be made clear why these were selected within the TRIM46 gene.

Response: We thank the reviewer’s suggestion. “We searched databases of NCBI RefSeq and Ensembl for annotated *Trim46* transcripts and found exon 8 and exon 10 are the two cassette exons consistently annotated in both databases. Importantly, skipping exon 10 causes a shift in the reading frame at the C-terminus, resulting in a much shorter isoform (TRIM46S) with an alternative stop codon in the last exon, exon 11 (Fig. 1e). We note the TRIM46 protein reported in previous studies is the long isoform (TRIM46L) including exon 10 and TRIM46S is an uncharacterized isoform^{4, 22}. The large difference between TRIM46L (83 kDa) and TRIM46S (60 kDa) suggests alternative splicing of exon 10 has functional implications.” (lines 142-149).

3. The same applies to the focus on PTBP2 as the only RNA-binding protein they analyze. Obviously it is established as an important brain-specific regulator, but were other candidate RBPs considered? Old CLIP data from the Darnell lab were used (Licatalosi, 2012; here: Figure

1i). At least in the supplement the entire gene should be shown in its iCLIP tag pattern, not only the exon 10 region. Since these old data were from whole mouse brain, CLIP should be repeated in their differentiation system to allow better comparison and correlation with the underlying pattern of alternative splicing changes.

Response: Reviewer #3 asked an important question. We did examine RNA-seq data of RBP KO brains, including PTBP2, RBFOX (GSE96722), NOVA (GSE69709), MBNL (GSE68890, GSE39911, GSE38497), and SRRM4 (GSE65818). We did not find significant splicing changes in these mutants except the *Ptbp2*-KO brain. PTBP2 was shown to coordinate the alternative splicing program during early axonogenesis, which increased our confidence to pursue this regulation. This does not mean PTBP2 is the only regulator or RBFOX/NOVA/MBNL/SRRM4 can't regulate *Trim46* in certain circumstances. We are now looking to identify additional upstream regulators of exon 8 and exon 10.

Regarding CLIP, we have included the CLIP browser track for the whole *Trim46* gene in Supplementary Fig. 3. We chose to use the CLIP data from the Darnell lab because the CLIP experiments were performed using embryonic day 18 (E18) mouse brains and we found that exon 10 splicing is altered in E18 *Ptbp2* KO brain (Fig. 1g and 1h). The combination of splicing changes and CLIP signals at the same embryonic stage of mouse brain provides strong evidence that exon 10 is regulated by PTBP2 *in vivo*. Over the years, we have been educated by many scientists in the neuroscience community to view *in vivo* data stronger evidence than *in vitro* data. We therefore did not revert to performing CLIP-seq in ESC-derived neurons. There is also a practical barrier of doing CLIP-Seq in ESC-derived neurons. A typical CLIP procedure requires >30 million neuronal cells as an input (ESC-derived neurons have lower RNA yield than regular cell lines). Our ESC differentiation procedure uses 96-well plates to form an embryonic body (EB) in individual wells. Meticulous care is needed to handle each EB. A whole 96-well plate of EBs can generate up to 1 million neurons if all goes well. Therefore, at least 30 counts of 96-well plates (or 2,880 wells) are needed for just one CLIP biological replicate. Finally, to achieve the Reviewer's goal that "CLIP should be repeated in their differentiation system to allow better comparison and correlation with the underlying pattern of alternative splicing changes", we need to create a *Ptbp2*^{-/-} ESC line and differentiate the *Ptbp2*^{-/-} line to neurons and further confirm exon 10 splicing change in *Ptbp2*^{-/-} neurons. All of these will take substantial efforts (more than a year) to complete, only to show what has already been observed in the mouse brain. Since the *in vivo* mouse data is already consistent with the *in vitro* differentiation model and that the *in vivo* mouse data shows exon 10 splicing in PTBP2 KO and PTBP2 CLIP-Seq at the same embryonic stage, we respectfully think the CLIP experiment in ESC-derived neurons is beyond the scope of the current manuscript.

4. Results, p.11 top line: Since at least in some experiments low residual levels of the short protein isoform can be detected (Figure 2j), "below detection sensitivity" should be changed.

Response: We cannot confidently prove the band is the TRIM46S protein. First, this band does not always show. Second, it migrates slightly faster than the expected 3xFLAG-TRIM46. Third, we know from our experience and many other experiments the FLAG antibody can detect a non-specific band around 60 kDa, though it is weak and requires longer exposure to show. We

tend to be more conservative when results are not well reproduced or uncertain. We have changed the wording to “the endogenous TRIM46S protein is difficult to detect with existing methods” on page 11, line 250-251.

5. Figure 3cd: Were these quantitations of protein levels after CHX treatment normalized?

Response: They were normalized. The FLAG-Trim46L and FLAG-Trim46S cDNA are followed by IRES-GFP in the CAGIG backbone vector. After transfection, FLAG-TRIM46 proteins were detected by the FLAG antibody and normalized to GFP first at each time point after CHX treatment and then to starting levels of the first collection time point (0 hour, i.e., right before CHX treatment). That is, the GFP blot was used as an internal control for each individual time point (0, 1, 3, 6, 12, 24 hr) as the first level of normalization (FLAG-TRIM46/GFP). Then all first level normalized values (FLAG-TRIM46/GFP) was divided by the corresponding value at 0 h as a second level of normalization to obtain “Protein Remaining (%)”. (lines 270-280)

6. Figures 5/6, exon 8 as an NMD target. This is well demonstrated by CHX treatment, as well as UPF2-KO and UPF1 siRNA-knockdown, and by conservation data. It would be even more convincing if another exon of TRIM46 were included as a negative control.

Response: This is a valid point. As suggested by the reviewer, we tested exon 10 under the two strongest NMD inhibition conditions: Emx1-Cre Upf2cKO cortices (in vivo), and CHX-treated primary cortical neurons (in vitro). “We found that in *Upf2*-cKO cortices the E8+ isoform was substantially upregulated at E14.5 (8.7-fold, $p=1.3 \times 10^{-5}$) and P1 (6.8-fold, $p=1.5 \times 10^{-5}$, Fig. 5e and 5f). By comparison, exon 10+ transcripts relative to exon 10- transcripts at E15.5 were marginally changed (1.5-fold, Supplementary Fig. 9c and 9d), probably under indirect effects of NMD inhibition. A similar small effect was observed in CHX-treated DIV 1 primary cortical neurons (1.4-fold, Supplementary Fig. 9e and 9f), in contrast to the much larger fold change observed in exon 8+ transcripts (5.0-fold, Fig. 5c and 5d)” (lines 399-406)

7. Did the authors make an attempt to identify what exon 10-dependent differences on the protein level and protein regions determine protein destabilization?

Response: Reviewer 3 asked a very interesting question. We did generate hypotheses and make some attempts to uncover the molecular underpinnings. TRIM46 belongs to the Tripartite motif (TRIM) protein family, which includes the RING (Really Interesting New Gene), B-box, and coiled coil domains. Given the RING domain is involved in ubiquitin-based protein degradation, we hypothesized TRIM46 carries an E3 ubiquitin ligase activity that may self-recognize and target TRIM46 proteins for degradation. In TRIM46L, the exon 10-encoded domain may function to block self-recognition or ubiquitination, whereas TRIM46S does not. A recent study (PMID: 32295675) shows that TRIM46 (without indicating which isoforms) binds PPAR- α directly and promotes PPAR- α polyubiquitination to induce ubiquitin-dependent degradation and to regulate cell cycle in osteosarcoma cells. The data indicate TRIM46 possesses a catalytic RING-E3 ubiquitin ligase activity. The exon 10 of TRIM46L encodes a SPRY domain that is typically involved in protein-protein interaction. This region may mask a degradation signal/motif of TRIM46L to prevent its rapid degradation and hence increase its protein stability. Without a similar SPRY motif, TRIM46S is vulnerable to proteasome and rapidly degraded (probably by

itself and/or by another E3 ubiquitin ligase). Supporting this idea, a reference database PhosphoSitePlus (v6.6.0.2) largely based on reported mass spectrometry analyses shows that the TRIM46 (presumably TRIM46S proteins since they are based on non-neural cell lines) is a highly ubiquitylated molecule at different motifs. We are still in the process of investigation and will need substantial amount of new data to make any conclusion. For this manuscript, we would like to stay focused on the coupling of multi-layered controls, how they influence TRIM46 expression and subsequent functional activity on axon formation. As shown, *Trim46* would be the first example where two alternative exons are coupled with two different downstream mechanisms and converge to influence the total gene output. Exon 8 and exon 10 splicing present double safeguards constituting a stringent mechanism to regulate the total TRIM46 protein level.

REVIEWERS' COMMENTS

Reviewer #1 (Remarks to the Author):

The authors have addressed my comments and significantly improved the paper!

Reviewer #2 (Remarks to the Author):

The authors have addressed my comments satisfactorily.

Reviewer #3 (Remarks to the Author):

December 17, 2021

Nature communications, NCOMMS-21-12859 A (revised manuscript)

Vuong et al., Zheng lab

(accept)

“Multilayered regulations of alternative splicing, NMD, and protein stability control temporal induction and tissue-specific expression of TRIM46 for axon formation”

The authors have thoroughly revised their manuscript, by addition of some more data in new figure panels and at multiple places in the Results section, also by modifying the title. The response to the minor comments from my side is very detailed and convincing, so that therefore I can accept this manuscript in its revised version. Very interesting and carefully done work.